# NMI and IFP35 serve as proinflammatory DAMPs during cellular infection and injury

Zhikai Xiahou[1,2], Xiangli Wang[1,2], Juan Shen[1,2], Xiaoxiao Zhu[3], Feng Xu[4], Rong Hu[5], Deyin Guo[6], Henan Li[7], Yong Tian [1,8], Yingfang Liu[1,6] & Huanhuan Liang [1]

Damage-associated molecular patterns (DAMP) trigger innate immune response and exacerbate inflammation to combat infection and cellular damage. Identifying DAMPs and revealing their functions are thus of crucial importance. Here we report that two molecules, N-myc and STAT interactor (NMI) and interferon-induced protein 35 (IFP35) act as DAMPs and are released by activated macrophages during lipopolysaccharide-induced septic shock or acetaminophen-induced liver injury. We show that extracellular NMI and IFP35 activate macrophages to release proinflammatory cytokines by activating nuclear factor-κB through the Toll-like receptor 4 pathway. In addition, the serum levels of NMI are increased in patients who succumbed to severe inflammation. NMI deficiency reduces inflammatory responses and mortality in mouse models of sepsis and liver injury. We therefore propose that extracellular NMI and IFP35 exacerbate inflammation as DAMPs, making them potential therapeutic targets for clinical intervention.

[1] National Laboratory of Biomacromolecules, Institute of Biophysics, Chinese Academy of Sciences, Beijing 100101, China. [2] College of Life Sciences, University of Chinese Academy of Sciences, Beijing 100049, China. [3] Laboratory Animal Center, Institute of Biophysics, Chinese Academy of Sciences, Beijing 100101, China. [4] Department of Infectious Diseases, Second Affiliated Hospital, Zhejiang University School of Medicine, Hangzhou 310058, China. [5] Center for Physical Examination, Beijing Anzhen Hospital, Capital Medical University, Beijing 100029, China. [6] School of Medicine, Sun Yat-Sen University, Guangzhou 510275, China. [7] Department of Clinical Laboratory, Peking University People's Hospital, Beijing 100044, China. [8] Key Laboratory of RNA Biology, Institute of Biophysics, Chinese Academy of Sciences, Beijing 100101, China. Zhikai Xiahou, Xiangli Wang and Juan Shen contributed equally to this work. Correspondence and requests for materials should be addressed to Y.L. (email: liuyingfang@ibp.ac.cn) or to H.L. (email: hhliang@moon.ibp.ac.cn)

Higher organisms have developed elaborate systems to control cellular stress induced by pathogen invasion or physical injury. Stressed cells release various cell-intrinsic factors to elicit tightly controlled immune responses ensuring effective elimination of affected cells without compromising the survival of the host. A group of factors known as damage-associated molecular pattern (DAMP) released upon tissue infection or injury are involved in the activation of tissue-adjacent phagocytes, ultimately resulting in the clearance of dysfunctional cells[1–3]. However, persistent inflammatory responses stimulated by dysregulated DAMPs cause organ destruction as a result of ensuing chronic and acute diseases, including atherosclerosis, arthritis, systemic inflammatory response syndrome, and sepsis[4–6]. Thus, it is important to identify and characterize DAMPs for their functions in inflammatory responses and potential for clinical treatment.

Some proinflammatory DAMPs have been identified, including uric acid[7], mitochondrial formyl peptides and DNA[8], histones[9], high-mobility group box protein 1 (HMGB1)[10], S100 calcium-binding proteins A8/A9[11, 12], heat shock proteins[13, 14], interleukin 33[15, 16] as well as cold-induced RNA-binding protein[17]. Most of the known DAMPs are recognized by cell surface pattern recognition receptors (PRR)[1, 2], including Toll-like receptors (TLR) and receptors for advanced glycation end products, most commonly expressed on macrophages and endothelial cells. Upon DAMP binding, PRRs trigger the activation of nuclear factor-κB (NF-κB), resulting in the acute expression and release of proinflammatory cytokines, such as tumor necrosis factor (TNF), interleukin 1β and interleukin 6 (IL-6). These cytokines then recruit granulocytes to the infected or injured tissues, stimulate lymphocyte differentiation, and promote inflammation[1–3]. Therefore, DAMPs contribute to the control of inflammatory responses, which are essential for providing protection and restoring homeostasis of the tissues.

During the investigation of various interferon (IFN)-stimulated genes of unknown functions, we identified two DAMPs, N-myc and STAT interactor (NMI) and IFN-induced protein 35 (IFP35), that belong to the IFP35 family[18–20]. Both NMI and IFP35 consist of a leucine zipper domain in their N-termini, followed by two tandem NMI/IFP35 domains (NID). While little is known about the function of IFP35[21], NMI acts as a transcriptional regulator in multiple signaling pathways in the nucleus. In response to interleukin 2 and IFN-γ, NMI interacts with signal transducer and activator of transcription (STAT) and enhances STATs-mediated transcription of downstream genes in the JAK–STAT pathway[22]. Together with breast cancer type 1 susceptibility protein, NMI downregulates c-Myc-induced transcription level of human telomerase reverse transcriptase gene[23]. In addition, NMI was reported to inhibit the proliferation and metastasis of cancer cells[23–26]. Upregulated by type I or type II IFNs, NMI and IFP35 assemble into high molecular mass complex mediated by their NID domains and further into speckle-like aggregations in the cytoplasm with unknown functions[27, 28]. Existing reports suggest that intracellular NMI and IFP35 are negative regulators of innate immune signaling pathways, targeting retinoic acid-inducible gene I, nuclear factor IFN regulatory factor 7 and NF-κB[29–31].

Different from these intracellular functions, here we investigate the extracellular functions of NMI and IFP35. Our results provide evidence that both proteins are actively released by macrophage to extracellular space during cell injury and pathogen invasion. These extracellular NMI and IFP35 activate the NF-κB signaling pathway of adjacent macrophages through TLR4 and promote their release of TNF, thus serving as proinflammatory DAMPs.

## Results

**Recombinant NMI induces inflammatory responses.** Initially, while generating antibodies against IFP35 family proteins, we noticed that intraperitoneal injection of a protein fragment of IFP35 resulted in the loss of all treated mice. Post-mortem examination indicated that these mice suffered from swollen abdomen and increased ascites, suggesting that their death was a result of inflammation. We speculated that our IFP35 protein fragment served as DAMP, hence inducing the acute inflammatory responses observed. Since such truncated protein is presumably absent in organisms, we wondered if the full-length IFP35 and NMI, the other member of the IFP35 family, displayed similar activities in vivo.

To assess their potential functions as DAMPs, we attempted to produce full-length recombinant proteins of IFP35 and NMI. Although we failed at producing IFP35, we obtained highly purified recombinant mouse NMI protein (mNMI) from *E. coil* (Supplementary Fig. 1a). To minimize potential contamination of our samples with lipopolysaccharides (LPS), known to be a potent stimulus of innate immune responses[32], we treated our purified protein sample with polymyxin B (PMB) beads. Using a Limulus amebocyte lysate assay, we showed that the LPS content of our protein samples was below the detection limit of this assay (<1 pg LPS per µg protein), around 3–4 orders of magnitude below the dose used for effective induction of inflammation.

To verify whether exposure of full-length NMI protein elicits inflammatory response in animals, $10 \text{ mg kg}^{-1}$ of mNMI protein was administered intravenously to healthy male mice (C57BL/6, $n = 5$). Within 24 h post-injection, the proinflammatory factors TNF and IL-6 were detected in the serum of mice at high levels, indicating that NMI induced an inflammatory response (Fig. 1a, b).

To test whether NMI stimulates the activation of macrophages in vitro, we monitored the expression and release of proinflammatory factors by mNMI treated macrophage cell lines. We found that TNF and IL-6 were released by mNMI treated mouse RAW264.7 cells (ATCC TIB-71™) in both dose-dependent and time-dependent manner (Fig. 1c-f). When mouse RAW264.7 cells ($2 \times 10^6$ cells $\text{ml}^{-1}$) were stimulated with $1 \text{ µg ml}^{-1}$ mNMI protein, ~1000 pg $\text{ml}^{-1}$ TNF was released 6 h post-stimulation. In addition, when stimulated with $1–10 \text{ µg ml}^{-1}$ mNMI, both mRNA and protein levels of TNF increased in a dose-dependent manner in mouse bone-marrow-derived macrophage (BMDM) cells. These levels were comparable to those measured in cells stimulated by $100 \text{ ng ml}^{-1}$ LPS (Supplementary Fig. 1b, c).

To test if the activity was caused by LPS contamination, we examined the activity of the purified mNMI protein using PMB, heat and trypsin treatment (Fig. 1g). When treated with PMB, which blocks the LPS-induced activity, ~95% of the activity of mNMI remained. In contrast, heat pre-treatment, which only slightly reduces the activity of LPS, significantly reduced the activity of mNMI protein to about 20%, tested using unpaired Student's *t*-test (**$P < 0.01$). In agreement with this, the activity of mNMI protein was destroyed by trypsin treatment. In addition, we purified recombinant human NMI protein (hNMI) from insect cells which is free of LPS (Supplementary Fig. 1a). As shown in Supplementary Fig. 1d, the hNMI protein from insect cells stimulated the release of TNF by THP1 cells (a human monocytic cell line, ATCC TIB-202™). These results clearly show that NMI protein intrinsically stimulates the inflammatory response of macrophages.

Most, if not all of the known DAMPs induce innate immune responses through triggering the activation of NF-κB[12, 14, 17, 33, 34]. Thus, we measured the activation of NF-κB following mNMI treatment. As phosphorylation of inhibitor of NF-κB (IκB) and nuclear translocation of p50/p65 are hallmarks of NF-κB activation, we examined the phosphorylation status of IκB and subcellular localization of p50/p65 using an immunoblotting method (Fig. 1h). In mouse Raw264.7 cells

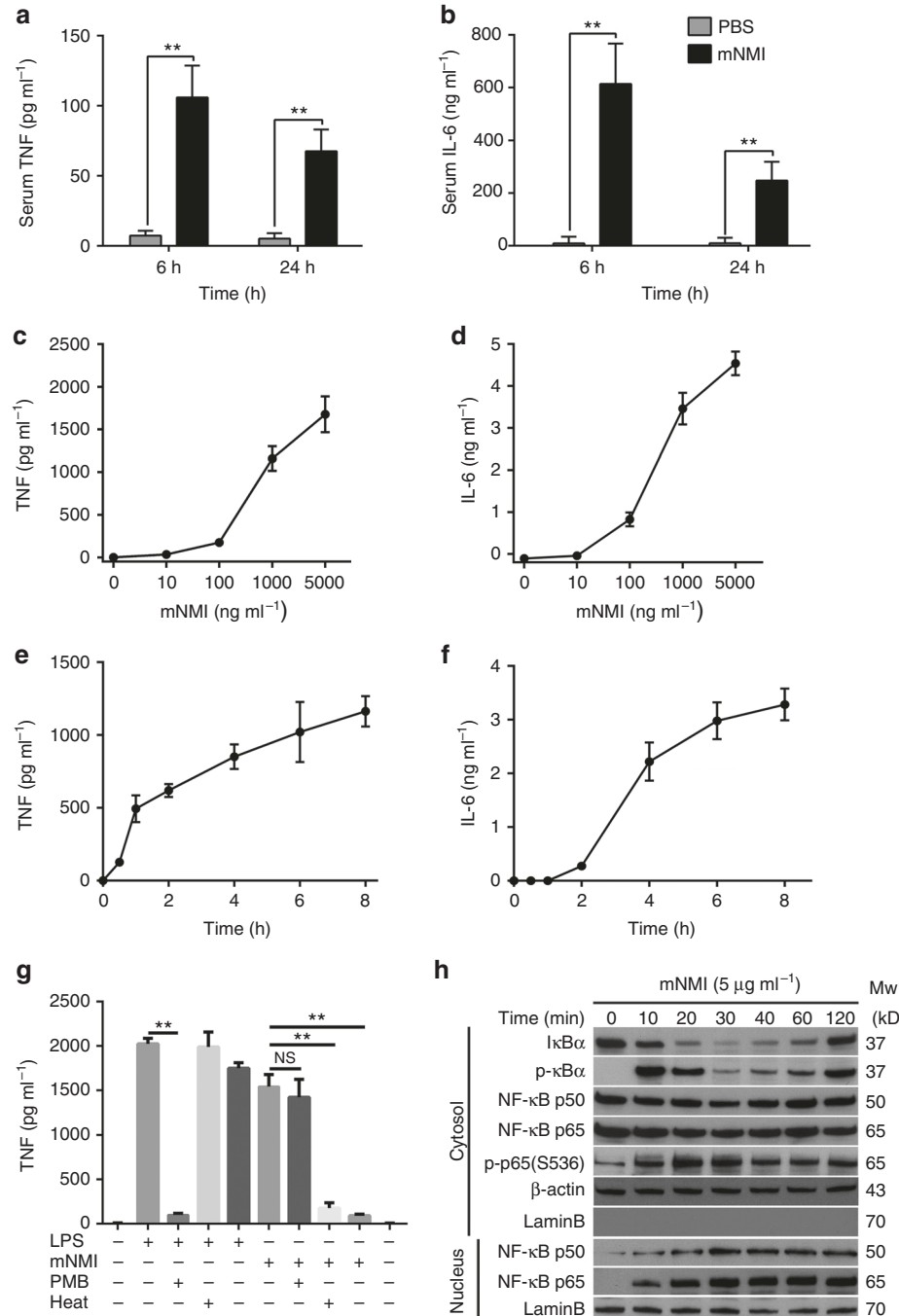

**Fig. 1** Recombinant NMI protein induces TNF release and NF-κB activation. **a**, **b** Tumor necrosis factor (TNF) and interleukin 6 (IL-6) in the sera of mice ($n = 5$) were determined by ELISA, 6 or 24 h after intravenous injection of recombinant mNMI (10 mg kg$^{-1}$) or PBS. **c**, **d** TNF and IL-6 released by mouse RAW264.7 cells (ATCC TIB-71™), 4 h post-stimulation with increasing concentrations of mNMI. **e**, **f** TNF and IL-6 released by RAW264.7 cells stimulated with 1 µg ml$^{-1}$ mNMI. **g** TNF released by RAW264.7 cells stimulated with mNMI (5 µg ml$^{-1}$) or LPS (100 ng ml$^{-1}$) for 8 h, with (+) or without (−) pre-treatment of polymyxin B (PMB) (25 µg ml$^{-1}$), heat (80 °C for 30 min) and trypsin (5 µg ml$^{-1}$, 37 °C overnight). **h** IκB, p-IκB, NF-κB (p50 and p65), phosphorated p65 at Ser536 (p-p65 (S536)) in the cytoplasm and NF-κB (p50 and p65) protein in nucleus extracts of RAW264.7 cells were assessed by immunoblotting, after the cells were treated with 5 µg ml$^{-1}$ recombinant mNMI protein. β-actin in the cytoplasm and LaminB in the nucleus were used as control. LaminB in the cytoplasm was detected to ensure that there was no contamination during the fractionation. Error bars in **c**–**g** indicate ± s.e.m. from three biological replicates. Significance was tested by unpaired Student's *t*-test test. *$P < 0.05$, **$P < 0.01$

(ATCC TIB-71™) treated with 5 µg ml$^{-1}$ recombinant mNMI protein, phosphorylated IκB (p-IκB) was observed within the first 10 min post-treatment, accompanied by p50/p65 translocation from the cytoplasm to the nucleus. This result supports the notion that NMI triggers inflammatory response through NF-κB activation.

**NMI is released by stimulated macrophages**. We next assessed whether NMI was produced and released into the media by activated macrophages. We treated human THP1 cells (ATCC TIB-202™) with a well-established inflammatory response stimulus, *Salmonella Typhimurium* (*S. typhimurium*). As shown in Fig. 2a, we detected NMI in the culture media within 60 min post-

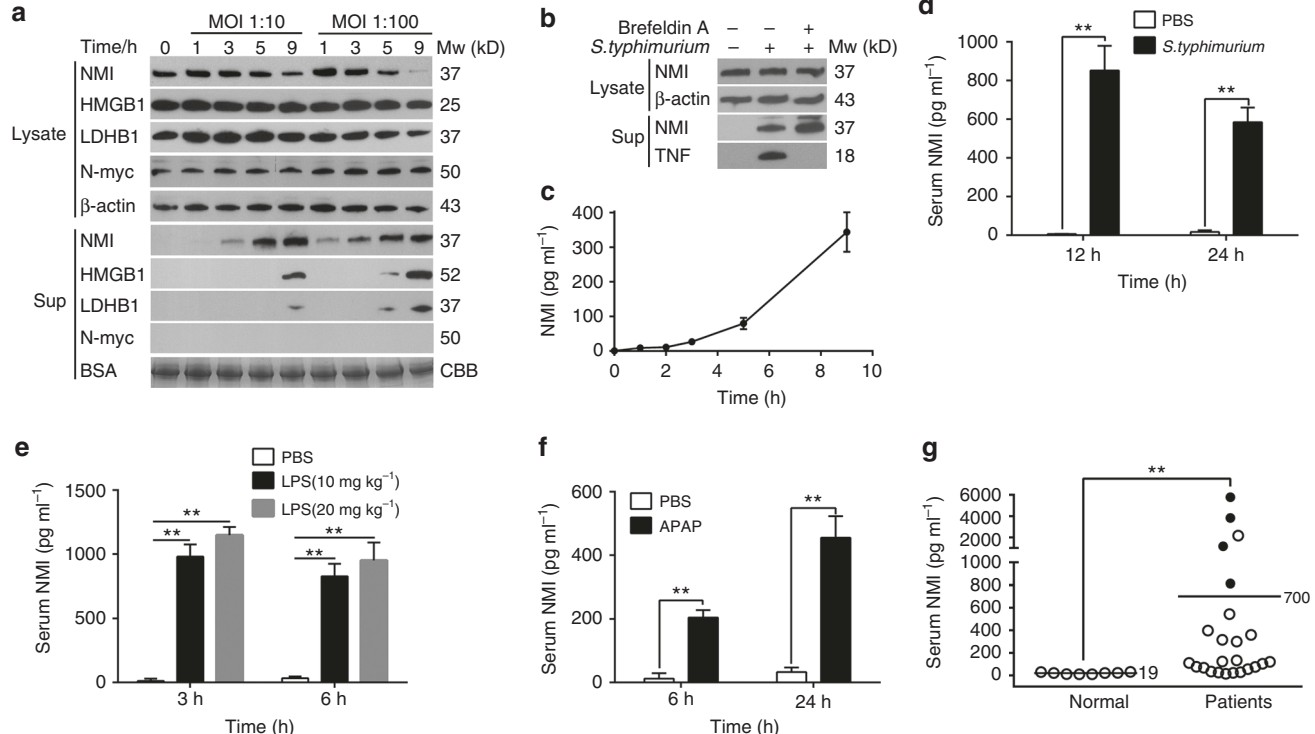

**Fig. 2** NMI is released by activated macrophages. **a** NMI in the supernatant (Sup) and cytoplasmic fraction of human THP1 cells (ATCC TIB-202™) were analyzed by immunoblotting post *S. typhimurium* stimulation, using high-mobility group box protein 1 (HMGB1), L-lactate dehydrogenase B 1 (LDHB1), and N-myc as control. β-actin in the cell and bovine serum albumin (BSA) in the supernatant were used to show equal loading. BSA was shown using commassie blue staining. **b** NMI in the supernatant (Sup) and cytoplasmic fraction of THP1 cells were analyzed by immunoblotting. The cells were pretreated with (+) or without (−) Brefeldin A for 8 h and stimulated with (+) or without (−) *S .typhimurium* for 4 h. **c** NMI released by mouse RAW264.7 cells (ATCC TIB-71™) ($2 \times 10^6$ cells ml$^{-1}$) was analyzed by ELISA at different time points post LPS (1 μg ml$^{-1}$) stimulation. Error bars indicate ± s.e.m. from three biological replicates. **d** Mice (*n* = 5 for each group) were intraperitoneally injected with $1 \times 10^4$ CFU live *S. typhimurium* per mouse and the concentration of the NMI in the serum were determined by ELISA. Data are presented as the mean ± s.e.m. **P < 0.01. **e, f** NMI in the sera of mice (*n* = 5 for each group) was determined by ELISA 3, 6 or 24 h after intraperitoneal injection of LPS (10 mg kg$^{-1}$ or 20 mg kg$^{-1}$) or acetaminophen (APAP) (300 mg kg$^{-1}$). Data are presented as the mean ± s.e.m. **P < 0.01. **g** ELISA analysis of NMI in the serum of 24 patients and 8 healthy individuals. Patients succumbed to severe inflammation were labeled with solid cycles. Significance in **d** and **e** were tested by unpaired Student's *t*-test. Significance in **g** was tested by Mann–Whitney *U*-test. **P < 0.01

infection. The extracellular NMI increased gradually in both a time-dependent and MOI-dependent manner. In contrast, N-myc, an NMI interact protein in nucleus, was not detected in the supernatant, suggesting that NMI was not released by damaged cells. Our results also show that the release of NMI occurs much earlier than the release of HMGB1, the known DAMP molecule. Given that *S. typhimurium*-induced pyroptosis occurred 6–9 h post-infection as indicated by the release of L-lactate dehydrogenase B 1 (LDHB1), an intracellular protein control, we propose that the extracellular NMI is actively released by stimulated macrophage. To test if NMI is secreted by classical endoplasmic reticulum (ER)–Golgi pathway, we employed Brefeldin A, an inhibitor of ER–Golgi vesicular secretion pathway. Our result shows the secretion of NMI is not affected by Brefeldin A (Fig. 2b), suggesting that NMI is not secreted by ER–Golgi vesicular secretion pathway.

To quantify the levels of extracellular NMI, we employed an in vitro enzyme-linked immunosorbent assay (ELISA), whereby mouse RAW264.7 cells (ATCC TIB-71™, $2 \times 10^6$ cells ml$^{-1}$) were stimulated with 1 μg ml$^{-1}$ LPS. mNMI release was detected 2 h post LPS-exposure, with mNMI levels increasing rapidly, by ~350 pg ml$^{-1}$ 9 h post LPS incubation (Fig. 2c).

To test whether NMI is released in vivo, we examined the release of NMI in animal models. First, we tested if NMI was released in acute inflammation responses induced by bacterial

infection. As shown in Fig. 2d, C57BL/6 mice were infected with *S. typhimurium* for 12 and 24 h (intraperitoneal injection of $1 \times 10^4$ CFU live *S. typhimurium* per mouse). mNMI was detected in the serum, with final levels of 500–1000 pg ml$^{-1}$. Next, we tested if NMI was released in sepsis, a life-threatening condition with organ dysfunction caused by a dysregulated host response to infection. A LPS-induced sepsis shock model was employed. In the mice model, the dendritic cells in the lymph nodes of the mice dramatically increased after the mice were exposure to median lethal doses of LPS (LD$_{100}$, 10 mg kg$^{-1}$) for 12 h, indicating their immune response was stimulated (Supplementary Fig. 2). As shown in Fig. 2e, the serum concentrations of mNMI in mice increased to ~1000 pg ml$^{-1}$ 3 h post LPS-injection (10 or 20 mg kg$^{-1}$). mNMI levels were found to be relatively stable between 3–6 h post LPS administration. In addition, to examine the function of NMI during organ failure in sterile inflammation condition, we induced liver injury in mice by administering 300 mg kg$^{-1}$ acetaminophen (APAP) (Fig. 2f). During the first 6 h, the concentration of mNMI reached ~200 pg ml$^{-1}$ level in the serum of mice, and increased to ~500 pg ml$^{-1}$ during the remaining 18 h of our time course. Together, these results show that NMI is released into blood circulation during infection or injury.

To test if NMI is released in individuals who encountered inflammation in clinical conditions, we analyzed the hNMI in the sera of 24 patients with severe inflammation caused by pathogens

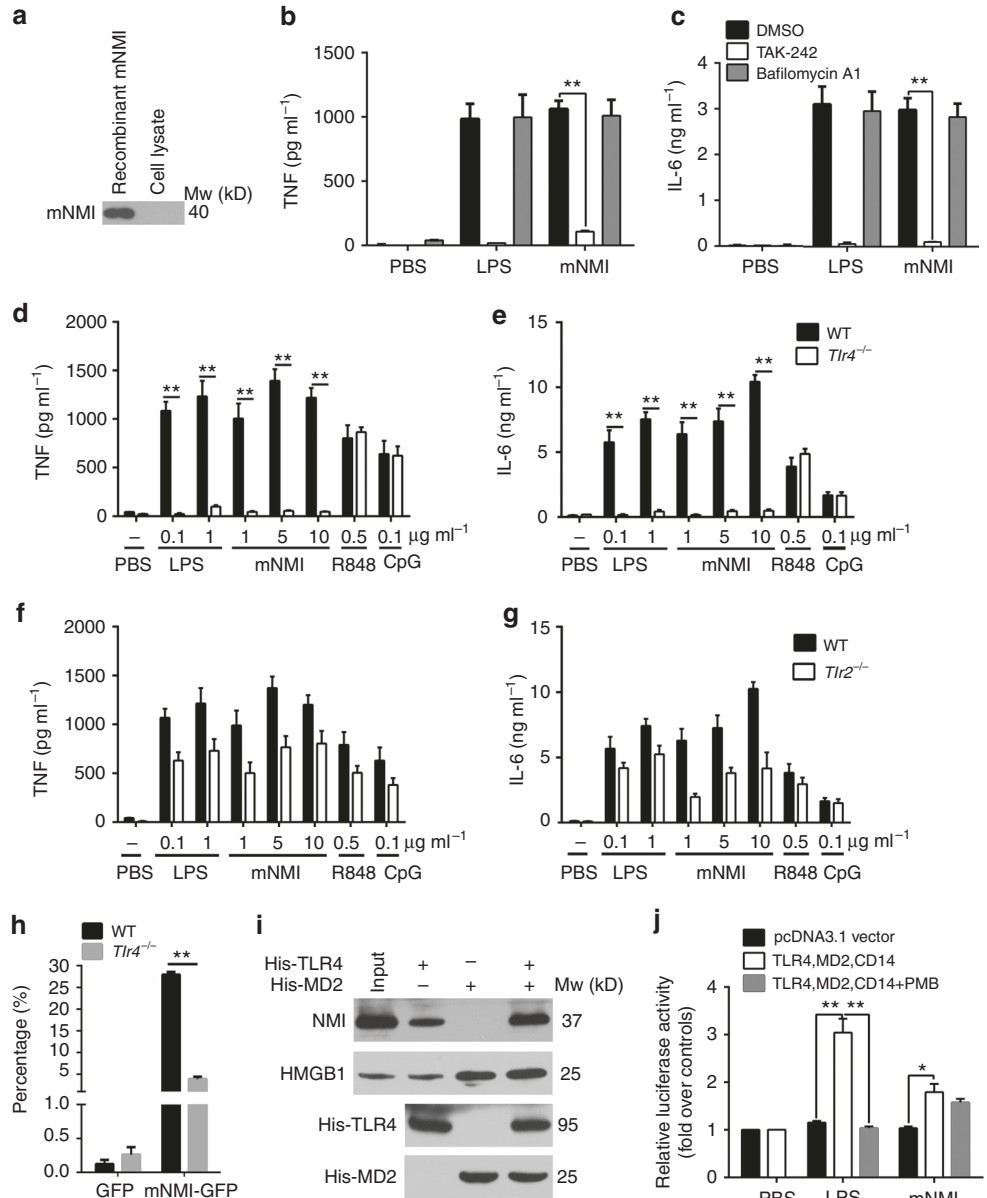

**Fig. 3** NMI stimulates macrophages through the TLR4 pathway. **a** Western blot analysis of mNMI in the BMDM cell lysate after 1 h incubation with recombinant mNMI. The BMDM cells were isolated from $Nmi^{-/-}$ mice and pretreated with macrophage colony-stimulating factor (MCSF). **b**, **c** TNF and IL-6 released by BMDMs from WT (C57BL/6) mice, pretreated with bafilomycin A1 (10 nM), TAK-242 (100 nM) or dimethyl sulphoxide (DMSO) for 2 h and stimulated with mNMI (5 μg ml$^{-1}$) or LPS (100 ng ml$^{-1}$) for 8 h. **d–g** TNF and IL-6 levels in the supernatants of BMDMs from WT, $Tlr4^{-/-}$ and $Tlr2^{-/-}$ mice were analyzed using ELISA 4 h post activation by different stimulus. **h** After incubation with mNMI-GFP for 1 h, the percentage of GFP labeled CD11b$^+$F4/80$^+$ cells was determined by Flow cytometric analysis. The cells were isolated from spleen in WT or $Tlr4^{-/-}$ mice. **i** NMI in the human THP1 cell (ATCC TIB-202™) lysates interacts with hTLR4. Ni-NTA beads coupled with 2 μg His-hTLR4 and/or His-hMD2 fusion proteins were used as bait. **j** The luciferase activity of HEK293T cells (ATCC CRL-11268™) are shown after stimulated with 5 μg ml$^{-1}$ mNMI for 4 h (in the presence or absence of 25 μg ml$^{-1}$ polymyxin B (PMB)). The cells were pre-transfected with mTLR4-MD2-CD14 and NF-κB promoter with luciferase activity. 100 ng ml$^{-1}$ LPS was administrated as positive control. In **b–e** and **g**, error bars indicate ± s.e.m. from 3 biological replicates. Significance was tested by one-way ANOVA followed by Student–Newman–Keuls test. **$P < 0.01$

infection using ELISA (Fig. 2g, Supplementary Table 1). Little hNMI was detected in the sera of the eight healthy participants. In contrast, serum hNMI levels were within a range of ~50–6000 pg ml$^{-1}$ in the patients. Such differences were probably due to the differences in inflammatory stages, pathogenic type and load, and physical conditions of the patients. Importantly, we found that patients with low hNMI serum levels survived and recovered after treatment. In contrast, 80% patients (four in five) with serum levels of 700 pg ml$^{-1}$ succumbed. These results suggest that serum

hNMI levels are correlated to the severity of inflammation and mortality of patients.

**NMI stimulates the activation of NF-κB through TLR4 pathway.** Since the intracellular NMI is involved in NF-κB regulation[30, 31], we examined if the secreted NMI was re-internalized into cells. The macrophage colony-stimulating factor treated $Nmi^{-/-}$ BMDM cells were incubated with mNMI

recombinant protein. After 1 h of incubation, mNMI was not detected in the cell lysate (Fig. 3a), suggesting that BMDM cells failed to take up mNMI. Thus, we propose that NMI stimulates NF-κB activation through membrane-based receptors.

To identify potential receptors involved in NMI-stimulated inflammatory response, we first examined the possible involvement of TLR, the receptors of several known DAMPs[2, 6]. When we used recombinant mNMI protein to stimulate BMDM cells, the release of TNF and IL-6 were found to be blocked by TAK-242, a TLR4 inhibitor (Fig. 3b, c). In contrast, bafilomycin A1, an inhibitor for endosome formation did not reduce the activity of mNMI. This result suggests that NMI stimulates the immune responses mainly through the TLR4 pathway. Consistent with these results, the activity of mNMI was blocked in $Tlr4^{-/-}$ BMDM cells (Fig. 3d, e). In contrast, in $Tlr2^{-/-}$ BMDM cells, the release of TNF and IL-6 were only slightly affected upon cell stimulation by recombinant mNMI protein (Fig. 3f, g). While this result was comparable to that of previous studies using LPS, a well-studied TLR4 ligand, it differed from that of studies using stimulation by R848 or CpG, two ligands of TLR7/8 and TLR9, respectively[35, 36]. The variation of the released proinflammatory factors between WT and $Tlr2^{-/-}$ stimulated by mNMI was comparable to the variation between that stimulated by LPS, R848, and CpG. Together, these results strongly indicate that NMI stimulates the immune responses mainly through the TLR4

pathway, but not through cell surface receptors TLR2 or its partners, TLR1 and TLR6[35], or endosomal TLRs 3, 7, 8, and 9.

We next examined the interaction between NMI and TLR4 on the surface of macrophages using Flow Cytometry. After incubated with mNMI–GFP, recombinant mNMI protein with green fluorescent protein (GFP) tag at its C-terminal, fluorescence signal was detected in 25–30% macrophage cells (Fig. 3h). In comparison, only 5% of $Tlr4^{-/-}$ macrophage cells incubated with GFP showed fluorescence signal. In order to test whether NMI and TLR4 physically interact with each other, we used a TLR4-based pull-down assay. As shown in Fig. 3i, the extracellular domain of human TLR4 protein (hTLR4 (24-631)) with His tag (His-TLR4) was used as bait to pull down NMI protein from cell lysate of human THP1 cells (ATCC TIB-202™). Different from HMGB1, the known TLR4–MD2 ligand, which is recruited by TLR4, MD2, and TLR4–MD2 complex respectively[37], NMI was recruited by TLR4 and TLR4–MD2, but not MD2 alone. The addition of MD2 further increased the binding ability of TLR4 to NMI.

In addition to TLR4 and MD2, CD14 is required for LPS-stimulated NF-κB activation[38, 39]. To examine whether the TLR4–MD2–CD14 complex is sufficient for NMI-stimulated immune responses, we employed a luciferase assay in combination with HEK293T cells (ATCC CRL -11268™) lacking this complex. In this assay, the HEK293T cells were transiently

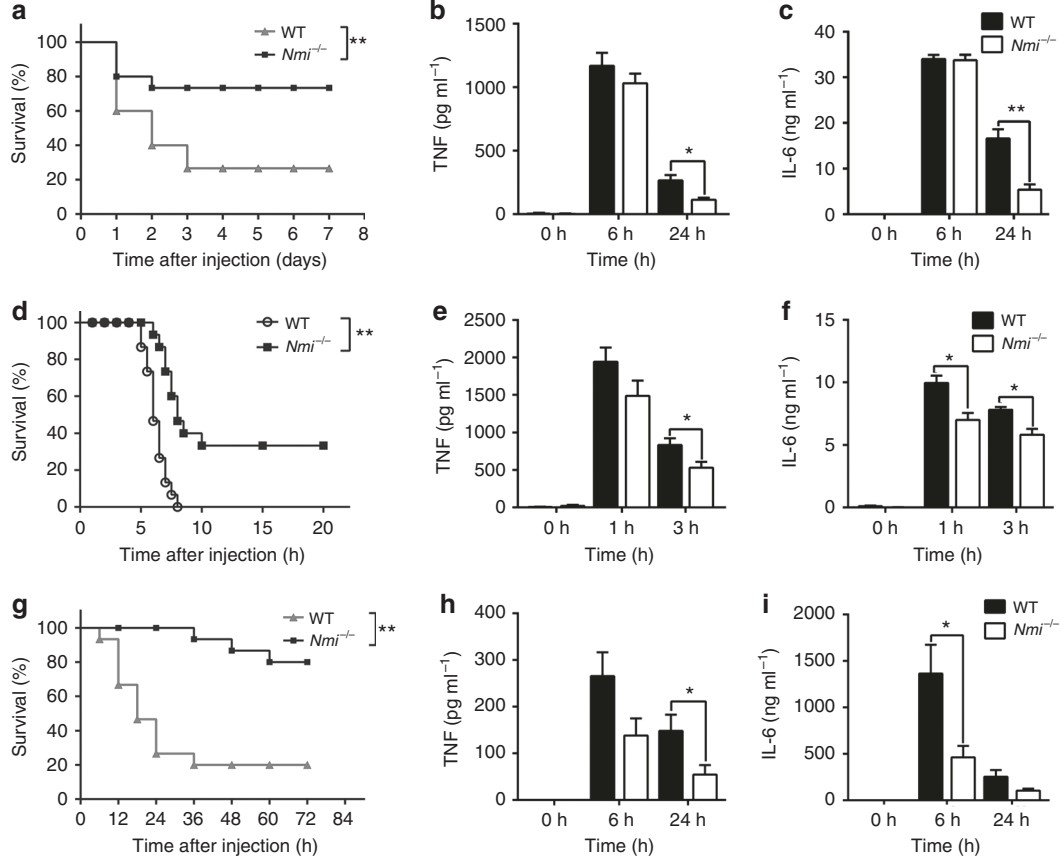

**Fig. 4** NMI knockout mice have attenuated inflammatory responses. **a** The survival rate of wild type (WT) (n = 15) and $Nmi^{-/-}$ mice (n = 15) intraperitoneally injected with LPS (50 mg kg$^{-1}$). *P < 0.05 compared to WT. **b, c** TNF and IL-6 in the sera of $Nmi^{-/-}$ and WT mice as determined by ELISA 6 or 24 h post LPS-injection. **d** The survival rate of WT(n = 14) and $Nmi^{-/-}$ mice (n = 14) intraperitoneally injected with LPS (50 μg kg$^{-1}$) and D-gal (1 g kg$^{-1}$). **P < 0.01. **e, f** TNF and IL-6 in the sera of $Nmi^{-/-}$ and WT mice as determined by ELISA 1 h or 3 h after injection of LPS and D-gal. **g** The survival rate of WT (n = 15) and $Nmi^{-/-}$ mice (n = 15) intraperitoneally injected with acetaminophen (APAP) (600 mg kg$^{-1}$). Significance in **a**, **d**, and **g** was determined using the log-rank test. **P < 0.01. **h, i** TNF and IL-6 in the sera of $Nmi^{-/-}$ and WT mice as determined by ELISA 6 or 24 h after injection of APAP. Error bars in **b**, **c**, **e**, **f**, **h**, **i** indicate ± s.e.m. from five individual mice. Significance was tested by unpaired Student's t-test. *P < 0.05, **P < 0.01

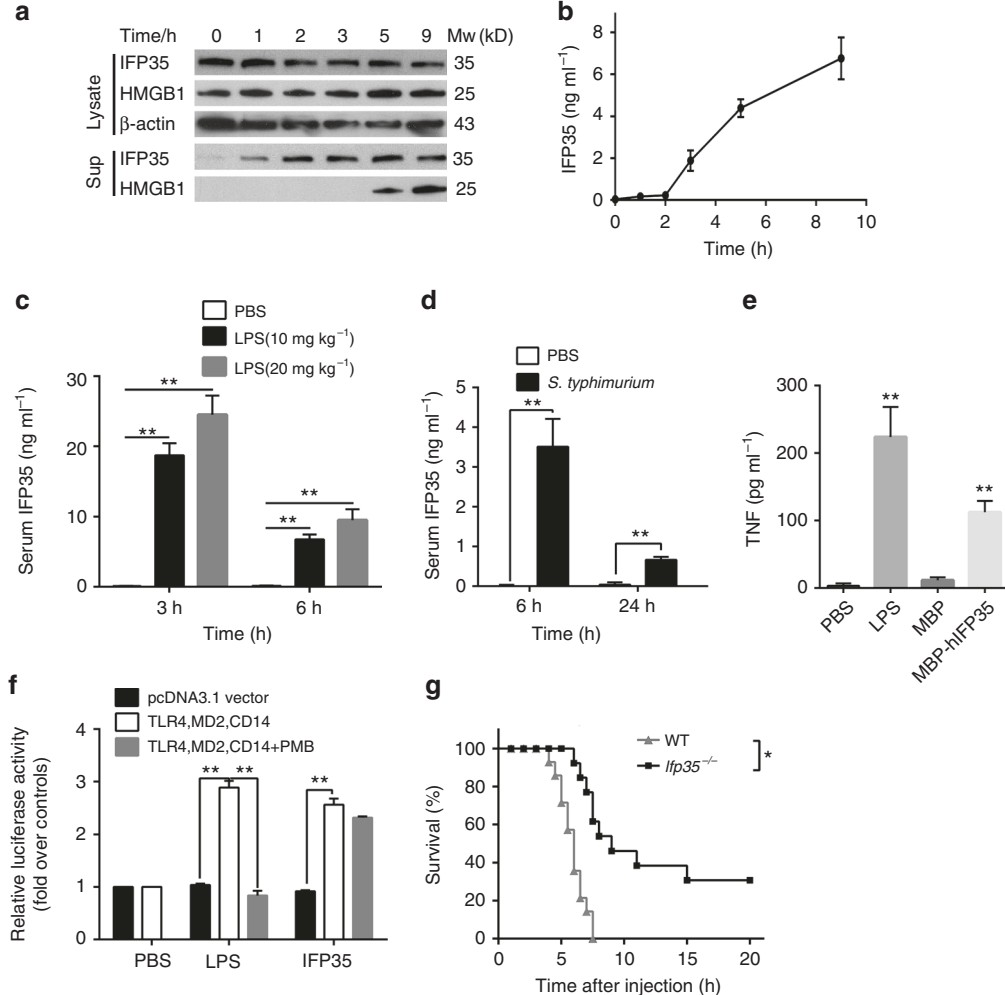

**Fig. 5** IFP35 serves as a DAMP and stimulates the inflammatory responses. **a**, **b** Immunoblotting **a** and ELISA analysis **b** of IFP35 in the supernatant and cytoplasmic fraction of mouse RAW264.7 cells (ATCC TIB-71™), treated with LPS (1 μg ml⁻¹). **c**, **d** Mice (n = 5 for each group) were intraperitoneally injected with LPS from *E. coli* (100 ng ml⁻¹, **c**) or live *S. typhimurium* (1 × 10⁴ CFU per mouse, **d**). The serum concentrations of IFP35 in mice were determined by ELISA. Data are presented as the mean ± s.e.m. Significance was tested by unpaired Student's *t*-test (\*\**P* < 0.01). **e** TNF level in the supernatants of human THP1 cells (ATCC TIB-202™) was analyzed by ELISA 8 h post incubation with LPS (100 ng ml⁻¹) or different purified proteins (5 μg ml⁻¹ respectively). Data are presented as the mean ± s.e.m of three independent experiments. \*\**P* < 0.01. **f** The luciferase activity of HEK293T cells (ATCC CRL-11268™), transiently transfected with the genes of hTLR4–MD2–CD14 and NF-κB promoter with luciferase, and stimulated with 5 μg ml⁻¹ hIFP35ΔN for 4 h (in the presence or absence of 25 μg ml⁻¹ polymyxin B (PMB)). 100 ng ml⁻¹ LPS was administrated as positive control. Data are presented as the mean ± s.e.m. of three independent transfections and are given as the ratio of luciferase activity relative to unstimulated cells. Significance in **d** and **e** was tested by one-way ANOVA followed by Student–Newman–Keuls test. \*\**P* < 0.01. **g** The survival rate of wild type (n = 14) and *Ifp35*⁻/⁻ mice (n = 14) intraperitoneally injected with LPS (50 μg kg⁻¹) and ᴅ-gal (1 g kg⁻¹). Significance was tested by log-rank test.\**P* < 0.05

transfected with plasmids expressing mTLR4, mMD2, mCD14, and NF-κB promoter-driven luciferase. 24 h post transfection, 5 μg ml⁻¹ of purified mNMI protein was added to the media for an additional 4 h incubation. We showed that recombinant mNMI protein triggered a 2-fold increase of luciferase activity compared with control (Fig. 3j). This activation was not affected by the LPS inhibitor PMB, indicating that the protein preparation was free of LPS. Together, these results show that NMI triggers NF-κB activation through the TLR4 pathway.

**NMI knockout attenuates inflammatory responses in mice.** To explore the possible application of NMI in clinical cases of sepsis, we investigated the function of NMI as DAMP using LPS-induced or *S. typhimurium*-induced murine sepsis models[12]. *Nmi*⁻/⁻ mice were generated using CRISPR–Cas9 technology[40, 41] (Supplementary Fig. 3a, b). Physiological or pathological

phenotype was not observed in the *Nmi*⁻/⁻ mice. The types and numbers of leukocytes in the peripheral blood, spleen, and lymph nodes are within normal range and comparable with those in WT, suggesting the normal development of the immune system in the knockout mice (Supplementary Fig. 3c, d). To set-up a survival model, lethal doses of LPS (30 mg kg⁻¹) were administered intraperitoneally to C57BL/6 WT and *Nmi*⁻/⁻ mice. Compared with a 20–30% survival rate of WT mice, more than 70% of the *Nmi*⁻/⁻ mice survived for 7 days post LPS-exposure (Fig. 4a). Because septic shock is mediated by proinflammatory cytokines, the serum concentrations of cytokines such as TNF and IL-6 after the administration of LPS were examined. Compared with WT, the release of toxic inflammatory factors in the serum of *Nmi*⁻/⁻ mice were reduced within the first 24 h post LPS-injection (Fig. 4b, c), suggesting that in the absence of NMI, the inflammatory responses in these mice were attenuated.

Resistance of $Nmi^{-/-}$ mice to endotoxic shock was confirmed by injecting mice with LPS in conjunction with D-galactosamine (D-gal) (Fig. 4d–f) or *S. typhimurium* (Supplementary Fig. 4a).

To assess the function of NMI in clinical cases of organ failure, we employed an APAP-induced hepatotoxicity model of mice[37]. As shown in Fig. 4g, wild-type animals showed symptoms of acute illness after receiving 600 mg kg$^{-1}$ APAP injection and 80% of mice died within the first 36 h. In contrast, ~80% of the $Nmi^{-/-}$ mice were still alive 72 h after APAP administration. Consistent with these results, serum levels of TNF, IL-6, aspartate transaminase (AST), and alanine transaminase (ALT) in $Nmi^{-/-}$ mice were also reduced (Fig. 4h, i and Supplementary Fig. 4b). Together, these findings support the hypothesis that high levels of NMI contribute to mortality in acute inflammation induced by infection or injury.

**Extracellular IFP35 serves as DAMP.** In order to investigate if IFP35 possesses DAMP characteristics, we first assessed if IFP35 was released in vitro by activated macrophages. Similar to NMI, IFP35 was released by mouse RAW264.7 cells (ATCC TIB-71™) following LPS-stimulation. The release time of IFP35 was earlier than that HMGB1 (Fig. 5a). The levels of extracellular IFP35 released by LPS-stimulated RAW264.7 cells were measured as 1–10 ng ml$^{-1}$ (Fig. 5b). In addition, IFP35 was also detected in the serum of septic mice induced by LPS or *S. typhimurium* (Fig. 5c, d).

Given the technical challenge in the production of recombinant IFP35 protein presents, we generated instead fused or truncated forms of human IFP35 protein (hIFP35). A hIFP35 fusion protein with maltose-binding protein (MBP) tag at its N-terminal (MBP–hIFP35) was expressed and purified form insect cells (Supplementary Fig. 5a). An N-terminally truncated form of hIFP35, termed hIFP35ΔN, including residues 31–289 of hIFP35, was expressed in *E. coil* (Supplementary Fig. 5a). Due to the high sequence conservation (identity >70%) between hIFP35 and mouse IFP35 (mIFP35), MBP–hIFP35, and hIFP35ΔN induced the release of TNF both from human THP1 (ATCC TIB-202™) and mouse RAW264.7 cells (ATCC TIB-71™) (Fig. 5e and Supplementary Fig. 5b). The activity of hIFP35ΔN was reduced by heat, destroyed by protease, not however, by PMB (Supplementary Fig. 5b). When purified hIFP35ΔN protein was added to HEK293T cells (ATCC CRL-11268™), it triggered the activation of the NF-κB pathway through pre-transfected hTLR4–MD2–CD14 complex, as detected by luciferase assay (Fig. 5f). In addition, $Ifp35^{-/-}$ mice were also generated using CRISPR–Cas9 technology[40, 41] (Supplementary Fig. 3). As shown in Fig. 5g, *Ifp35* knockout improved the survival rate of mice in sepsis models. For example, 50 μg kg$^{-1}$ LPS in conjunction with D-gal-induced lethality of WT mice was 100%. In comparison, the mortality of $Ifp35^{-/-}$ mice was ~70%. The levels of TNF and IL-6 in the sera of $Ifp35^{-/-}$ mice were reduced (Supplementary Fig. 5c). Together, these results suggest that IFP35 also functions as DAMP, thus contributes to the inflammatory responses upon infection.

## Discussion

Here, we present that extracellular NMI and IFP35 promote the innate immune response and acute inflammation as DAMP. Our results show that the IFP35 family members may be considered as promising targets in the clinical treatment of inflammation related diseases.

To investigate how NMI and IFP35 exacerbate the inflammatory response, we stimulated macrophages using recombinant NMI and IFP35 proteins under strict control. Our results showed that they activated NF-κB in macrophages through TLR 4 pathway, upregulated the transcription, and led to the release of proinflammatory factors including TNF and IL-6. The proinflammatory activities of NMI and IFP35 recombinant proteins are about 5–10% of that in LPS, the well-known virulence factor, but comparable to the activities of other DAMPs reported in previous studies[12, 17, 33]. Thus, NMI and IFP35 recombinant proteins may serve as immunopotentiators to improve the immune capacities in patients.

Different from the prototypical DAMPs which are released by pyroptotic cells, NMI and IFP35 were actively released by activated macrophages much earlier before cell damage occurred. In addition, we found that serum hNMI levels were directly correlated to the mortality of patients with severe inflammation. Thus, NMI and IFP35 may serve as early molecule marker for the diagnosis of sepsis. Furthermore, knockout of *Nmi* or *Ifp35* attenuate the inflammatory responses in septic mice and reduce their mortality, indicating that by blocking the extracellular IFP35 family proteins, it is possible to keep systemic or chronic inflammation under control[42, 43].

NMI was known as a transcriptional regulator in multiple signaling pathways. For example, recent results suggest that intracellular NMI and IFP35 proteins serve as negative regulators for IFN regulatory factor 7 and NF-κB pathways[29–31, 44]. Here, we found that the extracellular NMI and IFP35 promoted NF-κB activation. It is well accepted that proteins possess different functions in different locations. Since the extracellular NMI was not re-internalized into cells, we attribute the presence of such opposing functions of NMI and IFP35 to the fact that they probably play dual-functions in immune responses.

The functions of IFP35 family proteins were still incompletely understood. Lack of a recognizable secretion signal peptide is a common feature shared by IFP35 family proteins and some known DAMPs including HMGB1, S100 calcium-binding proteins, heat shock proteins, and interleukin 33, which make their secretion mechanisms elusive and unclear till now. Our results suggest that the release of NMI is independent on the classical ER–Golgi pathway or the inflammatory cell death pathway. In view of the various protein modifications and specific release pathways that may be present, the secretion mechanisms of NMI and IFP35 deserve further exploration. In addition, although both NMI and IFP35 trigger the activation of macrophage and exacerbate inflammation, the mechanisms of secretion and immune regulation of these two proteins may be different. The differences and possible crosstalk between IFP35 and NMI during inflammation need to be studied in the future.

Taken together, identification of NMI and IFP35 as DAMPs is important for a more complete understanding of the link between inflammation and immune responses. Our findings suggest considerable potential for NMI and IFP35 as useful targets for inflammation-induced diseases.

## Methods

**Plasmids.** Complementary DNA (cDNA) for mouse and human NMI, IFP35, TLR4, MD2, and CD14 were amplified by reverse transcription of mRNA from mouse RAW264.7 cells (ATCC TIB-71™) and human THP1 cells (ATCC TIB-202™), respectively. The *Mbp* and *Gfp* genes were amplified from pMAL-C2 vector (New England Biolabs, E8000S) and pEGFP-N1 vector (clontech, 6085-1), respectively. cDNAs of human or mouse TLR4, MD2, and CD14 were inserted into pcDNA3.1 vector (Invitrogen, V79020) for transient expression in HEK293T cells (ATCC CRL -11268™). The NF-κB promoter luciferase plasmid[45] was a gift from Dr Zhengfan Jiang (Peking University, Beijing). For recombinant expression, the DNA fragments of mNMI and mNMI-GFP was cloned into a modified pRSFDuet-1 vector (Novagen, 71341-3) with an N-terminal 6 × His tag, while the DNA fragment of hIFP35ΔN was inserted into a pGEX-6p-1 vector (Amersham, 27-4597-01) with an N-terminal glutathione S-transferase (GST) tag. The cDNAs of full-length hNMI and hIFP35 were cloned into pFastBac1 vector (Invitrogen, 10359016) which was modified to have an N-terminal MBP tag and PreScission Protease cleavage site. All plasmids were verified by DNA sequencing which was performed by BGI.

**Antibodies and ELISA kits**. Antibodies for NF-κB p50 (13586, 1:1000), p65 (8242, 1:1000), IκBα (4814, 1:1000), p-p65 (S536) (3033, 1:1000), p-IκBα (Ser32/36) (9246, 1:1000), anti-TNF (6945, 1:1000), and His-Tag (2365, 1:2000) were obtained from Cell Signalling Technology. Anti-hNMI (ab183724, 1:1000), anti-HMGB1 (ab79823, 1:1000), anti-LDHB1 (ab75167, 1:1000), anti-LaminB (ab133741, 1:1000), and anti-N-myc (ab24193, 1:500) antibodies were obtained from Abcam. Anti-β-actin (A5441, 1:10000) antibody was obtained from Sigma-Aldrich. Poly-clonal antibody for mNMI (Anti-mNMI, 1:1000) was prepared using recombinant mNMI protein according to previous report[46]. Anti-hIFP35 (H00003430-D01P, 1:1000) antibody was obtained from Abnova. The ELISA kits for hNMI (CSB-EL015893HU), mNMI (CSB-EL015893MO), mouse AST (CSB-E12649m), and mouse ALT (CSB-E16539m) were obtained from CUSABIO. The ELISA kits for mIFP35 (E10460m) was purchased from Wuhan EIAab Science Co. The ELISA kits for mouse IL-6 (431305) and mouse TNF (430905) were purchased from Biolegend.

The activity and specificity of anti-hNMI (ab183724), anti-mNMI, anti-hIFP35 (H00003430-D01P), mNMI (CSB-EL015893MO), and mIFP35 (E10460m) ELISA kits were tested as shown in Supplementary Fig. 6.

**Reagents**. Trichloroacetic acids (TCA; T6399-500), LPS from *E.coli* 055:B5 (L6529), D-gal (G0500-5G), and APAP (A0181-25G) were purchased from Sigma Aldrich. Macrophage colony-stimulating factor (MCSF) (15-02) was purchased from PeproTech. The CpG (ODN1826) was synthesized by BGI. The sequence of CpG is 5′-TCCATGACGTTCCTGACGTT-3′. The imidazoquinoline R848 (tlrl-r848), the TLR4 inhibitor TAK242 (tlrl-cli95) and the endosomal acidification inhibitor Bafilomycin A1 (tlrl-baf1) were purchased from InvivoGen. Recombinant mouse IFN-γ (485-MI-100) was purchased from R&D. Brefeldin A was purchased from Selleck (S7046).

**Cells and cell culture**. Mouse macrophage-like RAW264.7 cells (ATCC TIB-71™), human monocyte THP1 cells (ATCC TIB-202™) and HEK293T cells (ATCC CRL -11268™) were obtained from ATCC. Primary BMDM cells were isolated from WT (C57BL/6), *Tlr2*−/− and *Tlr4*−/− mice by following a standard procedure as pre-viously described[47]. The cell lines were tested to exclude the mycoplasma con-tamination using EEZ-PCR Mycoplasma Test Kit (BI, 20-700-20).

RAW264.7 macrophages were cultured in DMEM medium (Gibco, C11965500BT). THP1 cells were cultured in RPMI 1640 medium (Gibco, C11875500BT) with 0.05 mM β-mercaptoethanol (Gibco, 21985023) and primary BMDM cells were cultured in DMEM medium with 20 ng ml−1 MCSF (PeproTech, 315-02). All cultured media were supplemented with 10% heat-inactivated fetal bovine serum (FBS) (Gibco, 10099-141), 1% penicillin-streptomycin (Gibco, 15070063) and 2 mM glutamine (Gibco, 15140-122). Cells were maintained in a 37 °C incubator with 5% $CO_2$. All the cell lines have been tested for mycoplasma contamination by PCR.

For stimulation experiments, mouse RAW264.7 macrophages (ATCC TIB-71™), human THP1 cells (ATCC TIB-202™) and BMDM cells were seeded in 6-well plates at a density of $2 \times 10^6$ cells per well and grew overnight. Then the cells were stimulated with LPS (100 ng ml−1), *S. typhimurium* (SL1344), or varying concentrations of purified recombinant proteins mNMI, hNMI, hIFP35, MBP, MBP-hNMI, hIFP35ΔN, or MBP-hIFP35 for different times as indicated in the figures and figure legends.

Lymph node cells and spleen cells were obtained using the method as previously described[48]. Briefly, the regional axillary lymph nodes and mesenteric lymph nodes were taken from mice and cell suspensions were prepared by chopping the tissues into small pieces and gently grounding the pieces in RPMI-1640 medium (Gibco, C11875500BT) containing 10% heat-inactivated FBS (Gibco, 10099-141). Large tissue debris were removed by passing cell suspensions through a 40 μm cell strainer (BD Falcon, 352340), and then the cells were collected by centrifugation at $500 \times g$ for 5 min. Cell surface staining with blocking antibody and lineage-specific markers were performed after the cells were washed twice in RPMI-1640 medium. Spleen cell suspensions were prepared as described from lymph nodes, blood, and splenic red blood cells were lysed with Ack lysis buffer (Gibco, A1049201), incubated for 1 min at room temperature, then the spleen cells were centrifuged and washed twice in RPMI-1640 medium and stained. For quantification of total cells from lymph nodes and spleen, hemocytometer was used. With total cell numbers and the proportion of different cell lineages, the population of immune cells (including B cells, T cells, macrophages, dendritic cells and natural killer cells) in lymph nodes and spleen were calculated.

**Experimental animals**. All mice are on the C57BL/6 background. 8–12-week-old sex-matched mice were used unless described otherwise. All mice used were killed by cervical dislocation. C57BL/6 wild-type mice (000664) were purchased from Vital River Laboratory Animal Technology Co. *Tlr2*−/− (004650) and *Tlr4*−/− (003752) mice were from Jackson Laboratory.

The *Nmi* and *Ifp35* knockout mice were produced using CRISPR–Cas9 technology. The fertilized eggs were collected from the oviducts of superovulated female C57BL/6 mice mated to male C57BL/6 mice. Cas9 mRNA (150 ng ml−1) and transcribed sgRNA (100 ng ml−1) were mixed and microinjected into the cytoplasm of the fertilized eggs with well-recognized pronuclei in M2 medium

(Sigma, M7167-100ML). The sgRNA sequence for NMI is 5′-AAAACAAAGAACT AGACGAGG-3′, while the sgRNA sequence for IFP35 is 5′-CAGCTCAAAAGGG AGCGCACAGG-3′. The CRISPR technology generated frame shift to *Nmi* and *Ifp35* genes and resulted in the negative expression of NMI and IFP35. Approximately 100–250 zygotes were injected with each corresponding sgRNA and subsequently transferred to the uterus of pseudo-pregnant ICR females, from which viable founder mice were obtained.

All animals were maintained under specific pathogen-free conditions with approval by the institutional committee of the Institute of Biophysics, Chinese Academy of Sciences. All animal experiments were performed following the Ministry of Health national guidelines for housing and care of laboratory animals and were approved by the Institutional Animal Care and Use Committee (IACUC) at the Institute of Biophysics, Chinese Academy of Sciences for Medical Research.

**Human blood specimens**. Blood samples were obtained from the second Affiliated Hospital of Zhejiang University School of Medicine and some normal volunteers. Serum was separated and stored in aliquots at −80 °C. The informed consent was obtained from all participants and approved by the human research ethics com-mittee of the second Affiliated Hospital of Zhejiang University School of Medicine (NO. 016/2017).

**Recombinant proteins**. Recombinant human His-MD2, His-TLR4, and TLR4–MD2 complex were obtained from R&D Systems. Recombinant mNMI, mNMI–GFP, and GST-hIFP35ΔN protein were expressed in *E. coli*. The plasmid of mNMI, mNMI-GFP, or hIFP35ΔN was transformed into *E. coli* strain BL21 (DE3). Cells were cultured in Luria-Bertani medium at 37 °C with 100 mg l−1 ampicillin or kanamycin. When the $OD_{600}$ reached 0.8–1.0, the culture was induced by addition of isopropyl-β-D-thioglactosidase (Sigma, 11411446001) to a final concentration of 0.5 mM for 20 h at 16 °C. Cells were collected by cen-trifugation at 5000 rpm for 10 min. Pellets were resuspended in lysis buffer (20 mM Tris at pH 8.0, 400 mM NaCl) and lysed by sonication. For mNMI and mNMI-GFP purification, the lysate was separated by centrifugation at 16,000 rpm for 30 min, and the recovered supernatant was applied to a Ni-NTA affinity column (Qiagen), followed by intensive washing with washing buffer (20 mM Tris at pH 8.0, 150 mM NaCl, 50 mM imidazole). Recombinant protein was eluted from the Ni-NTA affinity column using elution buffer (20 mM Tris at pH 8.0, 150 mM NaCl, 500 mM imidazole) and further purified by gel filtration with a Superdex200 column (GE Healthcare) using lysis buffer as described above on an fast protein liquid chromatography protein purification system. To purify hIFP35ΔN, the lysate was separated by centrifugation at 16,000 rpm for 30 min and the supernatant was incubated with Glutathione Sepharose 4B resin (GE Healthcare) at 4 °C for 1 h. After incubation, the resin was washed with lysis buffer. The recombinant protein was eluted with 20 mM GSH and further purified by gel filtration with a Super-dex200 column (GE Healthcare) using lysis buffer as described above. The purity and integrity of recombinant proteins were verified by Coomassie blue staining after SDS–PAGE. Limulus amebocyte lysate assay was used to detect the potential LPS contamination.

The recombinant hNMI and hIFP35 were expressed using a Bac-to-bac expression system (Invitrogen). The baculoviruses encoding hNMI and hIFP35 were prepared using pFastBac1–MBP–hNMI and pFastBac1–MBP–hIFP35, respectively. Hi5 cells infected with the two viruses were cultured in ESF921 medium (Expression Systems, 96-001-01) for 60 h at 27 °C, respectively. Cells were harvested by centrifugation at 2500 rpm for 20 min. After centrifugation, the cell pellet was resuspended in lysis buffer (50 mM HEPES at pH 7.5, 400 mM NaCl, 500 mM Urea, and 5% glycerol) and lysed using cell homogenizer. The insoluble component of the cell lysate was removed by centrifugation at 16,000 rpm for 30 min. The recovered supernatants were loaded onto MBP columns (Novagen) which were pre-equilibrated with lysis buffer. After extensive washing with lysis buffer, target proteins were eluted with elution buffer (20 mM HEPES at pH 7.5, 400 mM NaCl, 30 mM maltose, and 5% glycerol). To purify MBP and hNMI, the eluted MBP–hNMI were cleaved by PPase and further purified by gel filtration with a Superdex200 column (GE Healthcare) using S200 buffer (20 mM HEPES at pH 7.5, 400 mM NaCl, 5% glycerol). To purify MBP–hNMI and MBP–hIFP35, the eluted MBP–hNMI and MBP–hIFP35 were directly purified by gel filtration with a Superdex200 column (GE Healthcare) using S200 buffer (20 mM HEPES at pH 7.5, 400 mM NaCl, 5% glycerol).

**RT-PCR**. The total RNA was extracted from HEK293T (ATCC CRL -11268™) and BMDM cells using Trizol reagent (InvivoGen, 15596018) according to the man-ufacturer's instructions. The first strand of cDNA was synthesized from 1 μg of total RNA using random primers and MMLV reverse transcriptase (Invitrogen, 28025-021). Real-time quantitative polymerase chain reaction analyses were per-formed using the CFX96 Real-Time PCR System (Bio-Rad). Glyceraldehyde-3-phosphate dehydrogenase (GAPDH) was used as an internal control for normal-ization, and the relative expression level of the analyzed gene was calculated by the ΔΔCt method. Each sample was measured in duplicates. The primer sequences are as follows: mGAPDH: sense, 5′-CAGAACATCATCCCTGCATC-3′; antisense, 5′-TACTTGGCAGGTTTCTCCAG-3′; mTNF: sense, 5′-CCAGTGTGGGAAGCT GTCTT-3′; antisense, 5′-AAGCAAAAGAGGAGGCAACA-3′.

**Pull down assay**. The THP1 cells (ATCC TIB-202™) were washed twice with cold phosphate-buffered saline (PBS) and collected in lysis buffer (20 mM Tris, pH7.5, 150 mM NaCl, 5 mM EDTA, 0.5% NP-40, 10% glycerol, protease inhibitor cocktail (Roche, 04693132001)). The whole-cell lysates were incubated at 4 °C for 45 min, followed by centrifugation (12,000 × g for 15 min, at 4 °C). Ni-NTA beads coupled with 2 μg His-TLR4 or His-MD2 fusion proteins were incubated with THP1 cell lysates at 4 °C for 4 h and then the beads were washed four times with lysis buffer. Proteins bound to the beads were separated by SDS–PAGE and immunoblotted with anti-His, anti-hNMI, and anti-HMGB1 antibody.

**Immunoblotting**. Human monocyte THP1 cells (ATCC TIB-202™) and RAW264.7 macrophages (ATCC TIB-71™) were pretreated with *S. typhimurium* (SL1344) or 100 ng ml⁻¹ LPS for 1, 3, 5, and 9 h. Then the cells were washed twice with cold PBS and collected in lysis buffer (20 mM Tris at pH 7.5, 150 mM NaCl, 5 mM EDTA, 0.5% NP-40, 10% glycerol, protease inhibitor cocktail (Roche, 04693132001)). The whole-cell lysates were incubated at 4 °C for 45 min, followed by centrifugation (12,000 × g for 15 min, at 4 °C). Secretory protein in the cell culture supernatant was collected by TCA/acetone precipitation. The cell culture supernatant was added 0.11 volumes of ice-cold 100% TCA and placed on ice for 2 h, followed by centrifuge at 20,000 × g for 30 min. Then carefully remove the supernatant, add 500 μl of acetone, centrifuge at 20,000 × g for 10 min, carefully remove the supernatant and dry the protein pellet in a vacuum evaporator. Protein samples were separated by SDS–PAGE, transferred to a PVDF membrane, and probed with specific antibodies against NMI, IFP35, HMGB1, LDHB1, and β-actin.

Mouse macrophage-like RAW264.7 cells (ATCC TIB-71™) were treated with purified recombinant protein mNMI for 10, 20, 30, 40, 60, and 120 min. Then the cells were collected in cold PBS, resuspended in hypotonic lysis buffer (10 mM HEPES at pH 8.0, 1.5 mM MgCl₂, 10 mM KCl, protease, and phosphatase inhibitors), and incubated on ice for 5 min. Cells were pelleted at 2000 × g for 3 min and the cytoplasmic fraction was removed to new tubes. The pellet was washed once with hypotonic buffer. The nuclear fraction was lysed in lysis buffer (20 mM Tris at pH 7.5, 150 mM NaCl, 5 mM EDTA, 0.5% NP-40, and protease inhibitors cocktail) on ice for 45 min. The lysate was clarified by centrifugation. NF-κB (p50 and p65), LaminB1, IkBα, p-IkBα, p-p65 (Ser536), and β-actin were detected by immunoblotting.

The uncropped western blots were shown in Supplementary Fig. 7.

**Luciferase assay**. HEK293T cells (ATCC CRL -11268™) were seeded in 6-well plates at a density of 2 × 10⁶ cells per well and grew overnight. The genes of TLR4, MD2, and CD14 were transiently transfected into HEK293T cells. 24 h after transfection, the NF-κB response element reporter was co-transfected with the Renilla reporter by using Fugene 6 (Promega, E2691). Cells were lysed with passive lysis buffer (Promega, E1941) and relative luciferase activity was determined by measuring fire flyluciferase activity and normalizing it to Renilla luciferase activity with the Dual Luciferase Reporter Assay System (Promega, E1910).

**Recombinant NMI induces proinflammatory cytokine release in mice**.
C57BL/6 male and age-matched (8–12 weeks of age) mice were used in the experiments. Mice (n = 5 for each group) were injected intravenously with recombinant mNMI (10 mg kg⁻¹) or PBS and serum concentrations of TNF and IL-6 in mice were determined by ELISA 6 and 24 h after injection.

**LPS-induced septic shock and APAP-induced liver injury in mice**. C57BL/6 male and age-matched (8–12 weeks of age) mice were used in the experiments. LPS from *E. coli* 055:B5 (Sigma-Aldrich, L6529) and APAP (Sigma-Aldrich, A0181-25G) were diluted in pyrogen-free saline. A combination of LPS (50 μg kg⁻¹) and D-gal (1 g kg⁻¹) were injected intraperitoneally in *Nmi⁻/⁻*, *Ifp35⁻/⁻*, and wild-type mice. Resistant of *Nmi⁻/⁻* mice to LPS-induced lethal toxicity were also observed when mice were injected with a large amount of LPS (50 mg kg⁻¹) without D-gal. For APAP-induced liver injury, APAP were injected intraperitoneally in *Nmi⁻/⁻* and wild-type mice. Observe and record the mortality of mice for 1 week.

***S. typhimurium*-induced abdominal sepsis**. C57BL/6 male and age-matched (8–12 weeks of age) mice were used in the experiments. In total, 1 × 10⁴ CFU *S. typhimurium* (SL1344) were diluted in pyrogen-free saline and injected intraperitoneally into *Nmi⁻/⁻* and wild-type mice. Observe and record the mortality of mice for 1 week.

**Flow cytometry**. Mouse cells from spleen and lymph nodes were incubated with anti-mouse CD16/CD32 (2.4G2, 1:200) in PBS containing 10% heat-inactivated FBS (Gibco) at 4 °C for 30 min, followed by staining with lineage-specific markers for 60 min under same condition; most of the antibodies were purchased from eBiosciences, while others were from BioLegend. For staining of B cells and T cells, we incubated cells with FITC-anti-CD3e (clone: 145-2C11, 1:400), APC-anti-CD19b (clone: eBiolD3, 1:200), PerCP/Cy5.5-anti-CD8a (clone: 53-6.7, 1:200), and PE-anti-CD4 (clone: GK1.5, 1:400). For macrophages, we incubated cells with FITC-anti-CD11b (clone: M1/70, 1:400) and PE-anti-F4/80 (clone: BM8, 1:200). For dendritic cells, cells were incubated with FITC-anti-CD11c (clone: N418,

1:400) and APC-anti-CD86 (clone: GL1, 1:200), and for natural killer cells, APC-anti-CD49b (clone: DX5, 1:100) were used. For the interaction between NMI and macrophages, cells isolated from spleen were incubated with GFP or mNMI-GFP (5 μg ml⁻¹) at 37 °C incubator for 1 h before stained with macrophage-specific markers as described above. After incubation, cells were analyzed by flow cytometry on BD FACS calibur with FlowJo (Tree Star).

**Statistical analysis**. Numerical data are expressed as the mean ± s.e.m. and were compared by unpaired Student's *t*-test or one-way ANOVA followed by Student–Newman–Keuls test. Differences between groups in animal survival experiments were determined using the log-rank test. Differences between NMI in the serum of patients and healthy individuals were analyzed using Mann–Whitney *U*-test. Some data sets had a statistical difference in the variation between groups. Differences in values were considered significant at *P < 0.05, **P < 0.01.

**Data availability**. All data that support the findings of this study are available from the corresponding authors upon reasonable request.

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

## Acknowledgements

We thank Bruce Beutler, Genhong Cheng, Xiaofeng Qin, and Bin Li for constructive discussion, Zhengfan Jiang, Feng Shao, Yangxin Fu, and Xiyun Yan for providing experimental materials, Torsten Juelich for suggestion and linguistic assistance during the preparation of the manuscript, Xiang Shi, Jing Cheng, Shu Meng, Xinyi Wu, Chengpeng Fu, and Lei Zhou for excellent animal technical support, Junying Jia and Shuang Sun for professional flow cytometry guidance. This work was supported by the NSFC foundation (31530015, 31470744, 31500609, 81572433), the Strategic Priority Research Program, CAS (XDB08000000), and the Frontier Sciences Project, CAS (QYZDY-SSW-SMC018).

## Author contributions

Z.X., X.W., and J.S. performed most of the experiments. Y.L. and J.S. found IFP35 was lethal to mice, which motivated the project. X.Z. and Y.T. generated the knockout mice. F.X. and R.H. collected samples from volunteers. D.G. and H.N.L. provided suggestions. H.H.L. initiated the project, designed the experiments and conceived the paper. Z.X., X.W., H.H.L., and Y.L. analyzed the data. H.H.L., Z.X., and X.W. wrote the paper, with revision from all co-authors. Y.L. directed the project.

## Additional information

**Competing interests:** The authors declare no competing financial interests.

