## [Peer Review File · Nature Communications]

Reviewers' comments:

Reviewer #1 (Remarks to the Author):

The authors find that NMI is a secreted protein, a novel observation that has never been seen before in the literature, and classify NMI as well as IFP35 as damage-associated molecular pattern molecules. Interestingly, they investigate the role of recombinant NMI and IFP35 on the inflammatory response of macrophages, and claim that both NMI and IFP35 are secreted by macrophages and are present in the sera of patients.

Mentioned below are various comments/critique of the manuscript:

Description of generation and characterization/confirmation/validation of multiple novel mouse models used in this study is lacking

- Validation of the Nmi and IFP35 KO mice is not presented. In addition, a detailed description of the strategy used to create the mice using CRISPR is lacking. Description of alterations made to the endogenous Nmi and IFP35 genes and confirmation that the products are non-functional is lacking.
- Characterization of the normal developmental phenotype of the innate immune system in the knockout mice is lacking.

Elucidation of the mechanism remains incomplete

- The paper does not present in vivo mobilization of immune populations to lymph nodes after injection of LPS into wild type and knockout mice. Serum TNF alpha and IL-6 are good initial indications of immune response however, physiologically they alone are not indicative of an immune response being mounted.
- The effect of Nmi on macrophage response in vivo needs to be assessed with a macrophage specific Cre to gain in vivo significance. The ubiquitous loss of Nmi could confound the mechanism as other innate immune cells are highly reliant on STAT signaling for function as well.

NF-kB as a mechanism for pro-inflammatory phenotype is incomplete and contradicts previous literature.

- The manuscript failed to address the previously published role of Nmi in NF-kB signaling and rule out NMI uptake into cells and actions besides binding TLR4 as the mechanism of NFkB stimulation. Thus this study contradicts previously published data showing that NMI inhibits NF-kB signaling activity through binding p65 and inhibiting acetylation by recruiting HDACs.

Insufficient experimentation to classify NMI and IFP35 as DAMPs

- This paper classifying NMI/IFP35 as a DAMP is unconventional in that it states that NMI as a DAMP is derived from monocytes while conventional DAMPs are produced from damaged cells other than immune cells.
- It is unclear how IFP35/NMI are secreted.

The manuscript comment that these proteins act to stimulate polarization of macrophages; however there is no data in this manuscript to support this claim

Figure 1.

- western blots of respective controls from each fraction are missing to show no cross contamination during the isolation.
- Loading controls are overexposed and unequal thus this will favor the trend that have been described in the manuscript and there is no evidence of NF-kB p65 subunit in the cytoplasm from any of these time points. Overall this result is incomplete.

1. Unclear if the BMDMs and RAW cells were pre-treated or co-treated with MCS-F. These would not be considered macrophages without initial MCS-F stimulation.

2. Figure 1 E lacks the corresponding blot for p-50 and p-65 in the cytosol. The manuscript claims that there is translocation from the cytoplasm to the nucleus but there is no evidence presented

Figure 2a. Needs to show equal loading of protein for the supernatant using commassie stain or other methods because all other results are reliant on commercially available ELISAs which are unvalidated. A positive control is required.

Figure 2 e describes the secretion of NMI in human patients with inflammatory responses. The authors make the claim that NMI is induced in patients with an active inflammatory response however most patients except two have levels around the low end at 50pg/ml. There needs to be more patients assessed to determine if this is a correct conclusion.

Figure 3 d describes a luciferase assay where in 4 plasmids are transfected and luciferase activity is assessed. It seems that transfecting 4 distinct plasmids would induce a large degree of variability given that there are some plasmids that may not be transfected at similar rates. It is unclear if there was any validation that these cells were expressing all these proteins prior to the luciferase assay.

Figure 3e. The authors would benefit from showing endogenous interaction of TLR4 and NMI to give physiological relevance to the mechanism identified in the paper.

- Murine Nmi antibody needs to be confirmed as the antibody has never been published before, therefore seeing as the manuscript relies heavily on this technique the specificity of the Ab needs to be tested and validated with the proper controls to ensure the ELISA results are valid.

Figure 4 describes knockout NMI essentially blocks inflammatory responses, however these experiments were done for 7 days which seems to be a more acute response. How does the loss of NMI effect more chronic inflammatory responses

Discussion is severely lacking in describing the importance of this work and clearly describing a role of IFP35 proteins in the inflammatory response.

Reviewer #2 (Remarks to the Author):

Xiahou et al show that IFP35 and NMI, two helix-loop-helix proteins, can be secreted by macrophage-like cells (presumably through an alternative pathway) upon activation with LPS, and serve as damage-associated molecular pattern molecules (DAMPs) by activating more macrophage-like cells via interaction with TLR4 and NF- κ B activation.

In general, the paper is well organized and fairly convincing.

I have however one fundamental criticism, and a few lesser concerns.

1-fundamental. DAMPs are generally cellular proteins that are either passively released by dead cells or actively secreted by activated cells (myeloid cell, but not only).

There is no attempt here to show that IFP35/NMI can be passively released by dead cells. This is important, for example, in pyroptosis, a form of cell death that depends on pore formation on the plasma membrane, followed by dispersal of intracellular proteins, for example IL-1 β and HMGB1 (the classical DAMP).

The authors should show that cells contain IFP35/NMI (presumably in the nucleus) before LPS

stimulation, and that the proteins become extracellular after induction of necrosis, for example freeze-thaw or sudden energy depletion (or pyroptosis, if the authors are inclined to execute a more complex experiment). Moreover, they should also show by microscopy that IFP35/NMI are translocated to the cytoplasm after cell exposure to LPS, and then to the extracellular medium, while (VERY IMPORTANT) LDH or other control extracellular proteins are not released. The LDH control is essential, otherwise one can argue that IFP35/NMI are released by dead cells only, and not actively secreted. One also needs microscopy to show that IFP35/NMI are NOT located in the ER-Golgi vesicular secretion pathway.

In fact, the authors state at line 139 of page 6 that "intracellular NMI decreased gradually in both a time- and MOI- dependent manner", but do not show the data. These data are essential.

2-important. The authors cannot say (page 5) that "heat pre-treatment... significantly destroyed the activity of NMI protein to about 20%", because destruction is a 100% reduction. In fact, an 80% decrease is not fully convincing, because it suggests that some of the protein refolds after heat treatment, or that the protein preparation contains pyrogens other than LPS (for example cell wall components). I suggest that the authors try digesting the protein preparation with trypsin; if some inflammatory activity remains, they should purify their protein better.

3-important. The role of caspase 1 is not well investigated, and actually looks minor. Casp1^{-/-} mice still secrete a lot of IFP35 (Fig 5b). This part can actually be omitted completely.

minor:

- in the introduction, the authors should explain what NMI/IFP35 are and what they do, and what is the phenotype of the Nmi knockout mouse (essential since they use it)
- the initial paper on secreted HMGB1, cited by the authors, used recombinant HMGB1 that was heavily contaminated with LPS. While this does not jeopardize the main conclusion of the paper, the size of the HMGB1 effects were overestimated. In practice, the authors should compare the effect of IFP35/NMI with that of HMGB1 reported in later papers (which is less; the effect of IFP35/NMI actually appears truly large).
- page 4, line 89: NMI is not homologous to IFP35, it may be paralogous (but this has to be argued). In practice, it is easier to say that NMI and IFP35 share xx% of sequence identity, or belong to the same protein family.
- the authors inappropriately use Student's test when they should use one-way ANOVA followed by post-tests between groups. This should not change the overall results, though.
- Western blots are very dark and often overexposed to saturation, and there is no mw marker.
- IFP35 and NMI can actually form heterodimers. Are the amounts of secreted IFP35 and NMI stoichiometric? Likewise NMI can form heterodimers with Myc. Is Myc secreted? Either yes or no, this would be interesting to know.

Reviewers' comments:

Reviewer #1 (Remarks to the Author):

The reviewer notes that the revised version of the manuscript reads significantly better and includes some valuable discussion points as well as improved overall flow. The reviewer requests the following items:

Answer to reviewer 1.2.1 – please include the flow cytometry plots of only dendritic cells recruited to the lymph nodes after LPS treatment.

Answer to reviewer 1.3.1 – please include discussion provided in the rebuttal in the main discussion section of the manuscript. This explanation of extracellular NMI having a distinct function to intracellular NMI is crucial to the mechanism of activation of macrophages by the NMI TLR4 interaction and needs to be clearly stated in the main body of discussion. The reviewer also suggests that data showing that NMI cannot be taken up into cells after secretion is an important overall finding and should be discussed in more detail in the results as well as discussion section of the manuscript.

Answer to reviewer 1.6.1 – please include the laminin blot for the cytoplasmic fraction and actin blot for the nuclear fraction to ensure that there was no contamination during the fractionation.

Answer to 1.8.3. – the reviewer appreciates the authors attempt to show specificity of their antibodies recognizing IFP35 and NMI however, using the knock out cells to validate an antibody when the knock out models themselves are also being validated using the same antibody is circular logic. Please overexpress the IFP35 and NMI cDNAs and confirm that the antibodies are specific utilizing the appropriate controls.

Reviewer #2 (Remarks to the Author):

The authors have replied to my queries and the other reviewers' in a complete and satisfactory way. Specifically, the absence of response to NMI and IFP35 proteins digested with trypsin indicates that there is little or no contamination from LPS and other PAMPs.

I still have some technical issues though.

Some Western blots are still overexposed to the point that they cannot be properly evaluated. The offending ones are in Fig 2a (the lysate part) and Fig 3e.

The statistical test used in Figure 2e should be indicated. Obviously, it must be a Mann-Whitney test. The correlation between patient outcome (dead/alive) and serum NMI appears stronger than the authors actually hint: the authors could apply a simple chi-square test (which turns out highly significant) or a more sophisticated ROC model.

Figure R5 should be added to the paper as a supplementary figure.

Figure R7 should be added as a new panel of Fig. 2 and appropriately described in the main text

Figure R8 should be added as a new panel to Fig 2 as a very convincing negative control.

Conversely, some supplementary figures should be shown in the main text or dispensed with. Fig S2b should be added as a new panel to Fig 2, Fig 2a can be omitted altogether. Fig S3a,b can be omitted altogether. The experiment in Fig S3c is not convincing, and should be omitted altogether. Fig S3d should be added as new panel to Fig 3, documenting physical interaction.

Minor: gene names for mice are indicated with the first letter capitalized and the subsequent ones in NOT capital

Reviewer #3 (Remarks to the Author):

They appear to have addressed most of the points that I made with new experimental data that on the face of it looks to be of good quality. The only substantive remark I have is that they should have had a control where the LPS was treated with protease (Fig. 1D, S6).

REVIEWERS' COMMENTS:

Reviewer #1 (Remarks to the Author):

This version of the manuscript is much superior to all the previous versions. Certainly, most issues that were raised have been addressed. However, the manuscript so far fails to clarify the following 3 things which this reviewer thinks are fundamentally germane to the premise and addition of these conceptual clarification will complete the manuscript in all aspects.

- The findings described state that secretion mechanism of NMI is ER-golgi independent (Figure 2b), it fails to elucidate which is the actual mechanism of secretion.
- Figure 2 states that NMI is released by activated macrophages and Figure 3 states that NMI stimulates macrophages. Thus, there appears to be a circular nature of signaling. Does this mean that the 1st secretion of these DAMPs occur through activated macrophages? If so, a diagrammatic representation of the proposed model of the action of these DAMPs will be highly desirable as a part of the published manuscript.
- Overall the manuscript has more details about NMI compared to IFP35. It is highly recommended to present additional supporting data is presented to make a more compelling case for IFP35 having the identical behavior. For example, it remains unclear if the secretion mechanism (or lack of it) discussed in Figure 2b is true about IFP35. It is also unclear if Figure 2 and Figure 3 are applicable to (extrapolated) statements made for IFP35. Additional data to support the claims of the manuscript that equate NMI and IFP 35 will fulfill this lacuna.

Reviewer #2 (Remarks to the Author):

The authors have satisfied all my queries, and I have no further comment. Congratulations for a well-done job.

Reviewer #3 (Remarks to the Author):

In this second revision the outstanding point I made has been addressed.

Point-by-point response to the reviewers' comments

We appreciate the reviewers for their critical and insightful comments, which greatly helped us to improve our research. To address these criticisms, we have carefully considered these comments, and provided further experimental data and analysis in the revised manuscript and supplementary information.

Reviewers' comments:

Reviewer #1 (Remarks to the Author):

The authors find that NMI is a secreted protein, a novel observation that has never been seen before in the literature, and classify NMI as well as IFP35 as damage-associated molecular pattern molecules. Interestingly, they investigate the role of recombinant NMI and IFP35 on the inflammatory response of macrophages, and claim that both NMI and IFP35 are secreted by macrophages and are present in the sera of patients.

Mentioned below are various comments/critique of the manuscript:

[Rev.1.1] Description of generation and characterization/confirmation/validation of multiple novel mouse models used in this study is lacking

[Rev.1.1.1]- Validation of the *Nmi* and *IFP35* KO mice is not presented. In addition, a detailed description of the strategy used to create the mice using CRISPR is lacking. Description of alterations made to the endogenous *Nmi* and *IFP35* genes and confirmation that the products are non-functional is lacking.

[Answer to Rev.1.1.1]: We thank you for the critical comments and apologize for lacking of these materials in the original manuscript. The description of the strategy to generate *NMI* and *IFP35* KO mice using CRISPR has now been included in the revised supplementary information. Briefly, the CRISPR technology generated both deletion and frame shift to *NMI* or *IFP35* genes. To evaluate these changes, we extracted the genome of the KO mice and amplified the *NMI* and *IFP35* genes. The alterations made to the endogenous *NMI* and *IFP35* genes by CRISPR were confirmed by electrophoresis and sequencing. CRISPR manipulation leads to failure of expression of endogenous *NMI* or *IFP35* in different organs, a result confirmed by Western Blot. All results are shown in Figure S4 a-b of the revised manuscript.

[Rev.1.1.2] - Characterization of the normal developmental phenotype of the innate immune system in the knockout mice is lacking.

[Answer to Rev.1.1.2]: To characterize the developmental phenotype of the innate immune system in our KO mice, we examined the leukocytes in the peripheral blood and the immune populations in spleen and lymph nodes. Our results showed that the types and numbers of these immune populations were within normal range and

comparable with those in WT, suggesting the normal development of the immune system in the KO mice. The results are shown in Figure S4 c-d in the revised manuscript.

[Rev.1.2] Elucidation of the mechanism remains incomplete

[Rev.1.2.1] - The paper does not present in vivo mobilization of immune populations to lymph nodes after injection of LPS into wild type and knockout mice. Serum TNF alpha and IL-6 are good initial indications of immune response. However, physiologically they alone are not indicative of an immune response being mounted.

[Answer to Rev.1.2.1]: As suggested by the reviewer, we have detected the mobilization of immune populations to lymph nodes. The results showed that the numbers of the dendritic cells in the lymph nodes dramatically increased 12 hours after injection of LPS into wild type and knockout mice, indicating an immune response are stimulated (Fig. R1). As LPS-induced immune response and sepsis mice model are already well established, we showed the serum levels of TNF α and IL-6 and the survival rates of the mice in Fig. 4 as those in previous reports^{1, 2, 3}. This data was not included in the revised manuscript.

Figure R1. Mobilization of immune populations to lymph nodes after injection of LPS into wild type and knockout mice. Data are presented as the mean \pm s.e.m. of 3 individual mice.

[Rev.1.2.2] - The effect of Nmi on macrophage response in vivo needs to be assessed with a macrophage specific Cre to gain in vivo significance. The ubiquitous loss of Nmi could confound the mechanism as other innate immune cells are highly reliant on STAT signaling for function as well.

[Answer to Rev.1.2.2]: We thank the reviewer for this suggestion. We have now performed additional experiments to assess the proinflammatory effect of NMI in macrophage-depleted mice injected with Clodronate (Cre) liposomes. Macrophage-depleted mice produced lower levels of serum TNF α than those measured in WT mice, indicating an attenuated immune response (Fig. R2). However, this result may come from the deficiency of NMI and other known DAMPs such as HMGB1 and S100A proteins, because they were also reported to be released by macrophages. In addition, although we showed that macrophage release NMI and IFP35, we cannot exclude the possibility that other immune cells possess this function as well. Therefore, we decided not to include this result in the revised manuscript.

Figure R2. Quantification of CD11b⁺F4/80⁺ cell populations in spleen after injection of Cre liposome or control neutral liposome is shown in the left panel. Serum concentrations of TNF α determined by ELISA 6 h after injection of recombinant mNMI are shown in the right panel.

Macrophage depletion. To induce macrophage depletion, Clophosome®-clodronate liposome (neutral) or control neutral liposome (FormuMax Scientific Inc., Palo Alto, CA, USA) were injected intraperitoneally into mice at the dose of 0.2 mL per mouse 24 h before NMI or PBS injection.

[Rev.1.3]NF- κ B as a mechanism for pro-inflammatory phenotype is incomplete and contradicts previous literature.

[Rev.1.3.1]- The manuscript failed to address the previously published role of Nmi in NF- κ B signaling and rule out NMI uptake into cells and actions besides binding

TLR4 as the mechanism of NF- κ B stimulation. Thus this study contradicts previously published data showing that NMI inhibits NF- κ B signaling activity through binding p65 and inhibiting acetylation by recruiting HDACs.

[Answer to Rev.1.3.1]: The reviewer points out the possibility that the secreted NMI might be re-internalized into cells and acts as the mechanism of NF- κ B stimulation. To address this concern, we tested internalization using MCSF-treated *NMI*^{-/-} BMDM cells. After 1 hour incubation with purified His-NMI, the cells were washed with PBS before Western Blot analysis. As shown in Fig. R3, His-NMI was not detected in the cell lysate (Fig. R3). This result suggests that cells failed to take up NMI. In addition, it is well accepted that proteins possess different functions in different locations. Thus, our results showed the extracellular function of NMI without contradicting previous reports describing the function of intracellular NMI. We attribute the presence of such opposing roles of NMI and IFP35 to the fact that they probably play a dual function in inflammatory responses.

Figure R3. Western blot analysis of His-NMI in the BMDM cell lysate after incubation with His-NMI. The BMDM cell were isolated from *NMI*^{-/-} mice and pretreated with MCSF.

[Rev.1.4] Insufficient experimentation to classify NMI and IFP35 as DAMPs

[Rev.1.4.1]- This paper classifying NMI/IFP35 as a DAMP is unconventional in that it states that NMI as a DAMP is derived from monocytes while conventional DAMPs are produced from damaged cells other than immune cells.

[Answer to Rev.1.4]: We thank you for the instructive comments. Most of the DAMPs are produced by damaged cells including HMGB1, histones, and mitochondrial DNA. However, an increasing number of studies showed that DAMPs are also produced by activated immune cells⁴. In the revised Fig. 2a, we added HMGB1 and an intracellular protein, L-lactate dehydrogenase B 1 (LDHB1), as control. We showed that NMI was released in 60 minutes post *S. typhimurium* infection. However, HMGB1 and LDHB1 were released 6-7 hours post infection.

Thus, we propose that the extracellular NMI was actively released by macrophages before damage.

[Rev.1.4.2] - It is unclear how IFP35/NMI are secreted.

[Answer to Rev.1.4.2]: Thank you for the insightful comments. The lack of a recognizable secretion signal peptide renders it difficult to understand the secretion mechanisms of NMI and IFP35. Here, we employed Brefeldin A, an ER-Golgi inhibitor to test if NMI was secreted via ER-Golgi vesicular secretion pathway. Our result showed that secretion of NMI failed to be blocked by Brefeldin A (Fig. R4). In addition, different from the prototypical DAMPs which are released by damage cells, the release of NMI and IFP35 could not be blocked by Caspase-1 knock out. In view of the various protein modifications and release pathways that may be present, future exploration of the secretion mechanisms for NMI/IFP35 will be necessary.

Figure R4: NMI in the supernatant (Sup) and cytoplasmic fraction of THP1 cells were analyzed by immunoblotting. The cells were pretreated with Brefeldin A for 8 hours and stimulated with *S.typhimurium* for 4 hours.

[Rev.1.5] The manuscript comment that these proteins act to stimulate polarization of macrophages; however there is no data in this manuscript to support this claim.

[Answer to Rev.1.5]: We apologize for the unclear description. Since macrophages stimulated by extracellular NMI release TNF α , the pro-inflammatory cytokine, they can be assumed to be in a pro-inflammation state. Thus, we hypothesized that NMI stimulate the polarization of macrophages towards the M1 state, the proinflammatory state in the original discussion. To make our discussion clearer, we decided to delete this hypothesis in the revised manuscript because this paper focuses on the DAMP function of NMI.

[Rev. 1.6] Figure 1.

[Answer to Rev.1.6.1] western blots of respective controls from each fraction are missing to show no cross contamination during the isolation. Loading controls are

overexposed and unequal thus this will favor the trend that have been described in the manuscript and there is no evidence of NF-kB p65 subunit in the cytoplasm from any of these time points. Overall this result is incomplete.

[Answer to Rev.1.6.1]: We thank you for the critical comments. We have now changed Figure 1e, and have included the revised loading controls (β -actin in the cytosol and Lamin B in the nucleus), and additional lanes showing p50, p65 levels in the cytoplasm. In addition, we tested the phosphate state of p65 using Anti-phosphate-p65 (S536).

[Rev.1.6.2] 1. Unclear if the BMDMs and RAW cells were pre-treated or co-treated with MCS-F. These would not be considered macrophages without initial MCS-F stimulation.

[Answer to Rev.1.6.2]: We apologize for the confusion caused by the insufficient description. BMDM cells were cultured in DMEM medium with 20 ng ml⁻¹ MCSF. RAW264.7 macrophages were cultured in DMEM medium as described in reference⁵. We have now included an additional sentence into the methods section of our revised manuscript.

[Rev.1.6.3] 2. Figure 1 E lacks the corresponding blot for p-50 and p-65 in the cytosol. The manuscript claims that there is translocation from the cytoplasm to the nucleus but there is no evidence presented.

[Answer to Rev.1.6.2]: We apologize for this omission, and have now included the loading controls and the p-50 and p-65 in the cytosol, as shown in the revised Fig.1e.

[Rev.1.7.1] Figure 2a. Needs to show equal loading of protein for the supernatant using commassie stain or other methods because all other results are reliant on commercially available ELISAs which are unvalidated. A positive control is required.

[Answer to Rev.1.7.1]: We thank you for this helpful suggestion. We have now added a loading control and two positive controls. The commassie stained BSA protein in the supernatant was using as loading controls. Late inflammatory cytokine HMGB1 and intracellular protein LDHB1 were used as positive control. All the new data were included in the revised Fig. 2a.

[Rev.1.7.2] Figure 2e describes the secretion of NMI in human patients with inflammatory responses. The authors make the claim that NMI is induced in patients

with an active inflammatory response however most patients except two have levels around the low end at 50pg/ml. There needs to be more patients assessed to determine if this is a correct conclusion.

[Answer to Rev.1.7.2]: In addition to the 8 patients in the previous manuscript, we identified an additional 16 patients, and incorporated those data into the revised Fig. 2e and Table S1. Our new results clearly show that serum NMI levels are correlated with mortality of these patients. Patients with low NMI serum levels survived and recovered after treatment. In contrast, 80% patients with serum levels of 700 pg ml⁻¹ NMI succumbed to sepsis.

[Rev.1.8.1] Figure 3d describes a luciferase assay where in 4 plasmids are transfected and luciferase activity is assessed. It seems that transfecting 4 distinct plasmids would induce a large degree of variability given that there are some plasmids that may not be transfected at similar rates. It is unclear if there was any validation that these cells were expressing all these proteins prior to the luciferase assay.

[Answer to Rev.1.8.1] The luciferase assay was set up according to reference³. In HEK293 cells, CD14 is endogenously expressed. However, TLR4 and MD2 are not expressed prior to the luciferase assay. As suggested by the reviewer, we checked the expression levels of these three proteins and found that they were expressed at similar level after the cells were transfected by the plasmids. These results are shown in the new Figure. S3b in the revised manuscript.

[Rev.1.8.2] Figure 3e. The authors would benefit from showing endogenous interaction of TLR4 and NMI to give physiological relevance to the mechanism identified in the paper.

[Answer to Rev.1.8.2]: We thank the reviewer for this invaluable suggestion. However, due to the low levels of released NMI, we failed to detect any interaction of endogenous NMI with TLR4 on the surface of macrophages. To circumvent these experimental challenges, we generated a mNMI-GFP recombinant protein, with the GFP-tag located at the mNMI C-terminal. To detect any interaction between mNMI-GFP and endogenous TLR4, we used Flow Cytometry. After incubating with mNMI-GFP, 25-30% macrophage cells were labeled in mNMI-GFP. In contrast, the percentage decreased to 5% using *TLR4*^{-/-} macrophage cells. This result is provided in a new Fig. S3d. Together with our results from the pull down assay (Fig. 3e), we provide strong evidence that NMI interacts with TLR4 *in vivo*.

[Rev. 1.8.3] - Murine Nmi antibody needs to be confirmed as the antibody has never been published before, therefore seeing as the manuscript relies heavily on this technique the specificity of the Ab needs to be tested and validated with the proper controls to ensure the ELISA results are valid.

[Answer to Rev.1.8.3]: Thank you for the critical comments. We evaluated the specificity of the mNMI or mIFP35 ELISA kits using *mNMI*^{-/-} or *mIFP35*^{-/-} cells as negative controls. As shown in Fig. 2b and 5a, the ELISA Kits respond to the culture of LPS stimulated WT cells. However, they do not respond to the culture of *mNMI*^{-/-} or *mIFP35*^{-/-} cells after LPS stimulation (Figure R5a). In addition, the protein interacted with the antibodies in mNMI or mIFP35 ELISA Kits were confirmed using Pull-down assay and Western blot (Figure R5b).

Figure R5: Pull-down assay was used to confirm the antibodies in mIFP35 and mNMI Elisa kits. (a) ELISA analysis of IFP35 and NMI in the supernatant of WT, *mNMI*^{-/-} or *mIFP35*^{-/-} BMDM cells, treated with LPS (1 $\mu\text{g ml}^{-1}$). (b) The WT, *mNMI*^{-/-} or *mIFP35*^{-/-} BMDM cells were washed twice with cold PBS buffer and collected in lysis buffer. The whole cell lysates were incubated at 4°C for 45 min, followed by centrifugation (12 000 $\times g$ for 15 min, at 4°C). mIFP35 or mNMI antibody from the Elisa kits was coupled with Protein G beads. The beads with antibody were incubated with cell lysates at 4°C for 4 hours and washed 4 times with lysis buffer. Proteins bound to the beads were separated by SDS-PAGE and immunoblotted with anti-IFP35 or mNMI antibodies.

Similarly, the protein interacted with the antibodies in hNMI ELISA Kits was

confirmed using Pull-down assay and mass spectrum. The top three of the mass spectrum results are listed as following:

Rank	Accession	Description	Score	Coverage	Unique Peptides
1	P02768	Serum albumin OS=Homo sapiens GN=ALB PE=1 SV=2 - [ALBU_HUMAN]	134.66	70.94	33
2	Q13287	N-myc-interactor OS=Homo sapiens GN=NMI PE=1 SV=2 - [NMI_HUMAN]	57.96	60.26	14
3	P01834	Ig kappa chain C region OS=Homo sapiens GN=IGKC PE=1 SV=1 - [IGKC_HUMAN]	31.40	80.19	6

[Rev. 1.8.4] Figure 4 describes knockout NMI essentially blocks inflammatory responses, however these experiments were done for 7 days which seems to be a more acute response. How does the loss of NMI effect more chronic inflammatory responses

[Answer to Rev. 1.8.4]: We thank the reviewer for this insightful question. We are currently setting up a model to study chronic responses, however, we would like to argue that those studies are beyond the scope of this manuscript, in which we focused on the acute immune responses induced by NMI.

[Rev. 1.8.4] Discussion is severely lacking in describing the importance of this work and clearly describing a role of IFP35 proteins in the inflammatory response.

[Answer to Rev. 1.8.4]: We have now rewritten the discussion to discuss in more detail the relevance of our findings.

Reviewer #2 (Remarks to the Author):

Xiahou et al show that IFP35 and NMI, two helix-loop-helix proteins, can be secreted by macrophage-like cells (presumably through an alternative pathway) upon activation with LPS, and serve as damage-associate molecular pattern molecules (DAMPs) by activating more macrophages-like cells via interaction with TLR4 and

NF- κ B activation.

In general, the paper is well organized and fairly convincing.

I have however one fundamental criticism, and a few lesser concerns.

[Rev. 2.1.1] 1-fundamental. DAMPs are generally cellular proteins that are either passively released by dead cells or actively secreted by activated cells (myeloid cell, but not only).

There is no attempt here to show that IFP35/NMI can be passively released by dead cells. This is important, for example, in pyroptosis, a form of cell death that depends on pore formation on the plasma membrane, followed by dispersal of intracellular proteins, for example IL-1 β and HMGB1 (the classical DAMP). The authors should show that cells contain IFP35/NMI (presumably in the nucleus) before LPS stimulation, and that the proteins become extracellular after induction of necrosis, for example freeze-thaw or sudden energy depletion (or pyroptosis, if the authors are inclined to execute a more complex experiment).

[Answer to Rev.2.1.1]: We thank the reviewer for the positive comments, and the invaluable suggestions on how to improve our arguments. To test if NMI is passively released by dead cells, we tested the translocation of NMI before or after cell pyroptosis caused by sudden energy depletion (Figure R6). The intracellular protein LDHB1 (L-lactate dehydrogenase B 1) is released by dead cells and used as control for pyroptosis. Our results show that the release of NMI started 6 hours post cell starvation, much earlier than the release of HMGB1 and LDHB1, which began at the 24-48 hours after starvation. These findings strongly suggest that NMI is first actively secreted, before it is passively released by dead cells. When our cells were infected by *S. typhimurium*, similar results were obtained. We have now incorporated these new results into Fig. 2a of the revised manuscript.

Figure R6. NMI, HMGB1 and LDHB1 in the supernatant (Sup) and cytoplasmic fraction of THP1 cells were analyzed by immunoblotting post starvation.

[Rev. 2.1.2] Moreover, they should also show by microscopy that IFP35/NMI are translocated to the cytoplasm after cell exposure to LPS, and then to the extracellular medium, while (VERY IMPORTANT) LDH or other control extracellular proteins are not released. The LDH control is essential, otherwise one can argue that IFP35/NMI are released by dead cells only, and not actively secreted. One also needs microscopy to show that IFP35/NMI are NOT located in the ER-Golgi vesicular secretion pathway.

[Answer to Rev. 2.1.2]: In our hands, endogenous NMI localized in the cytoplasm (Figure R7, left panel), which is consistent with the previous report^{6, 7}. After cell exposure to LPS, NMI and IFP35 translocated to the extracellular space. Since it is not easy to detect secreted NMI/IFP35 in culture using microscopy, the translocation of NMI was detected by Western blot using LDHB1 as control. Our result showed that NMI appeared in the culture media within 60 minutes post *S. typhimurium* infection, while LDHB1 was not released. The result is shown in the revised Fig. 2a. Thus, we propose that NMI is actively released.

As shown in Figure R7, NMI spreads in the cytoplasm of THP1 cells. It is hard to distinguish if it is located in the ER-Golgi vesicular secretion pathway by microscopy. Thus, to test if NMI is secreted by ER-Golgi vesicular secretion pathway, we employed Brefeldin A, an ER-Golgi pathway inhibitor. Our result show the secretion of NMI is not affected by Brefeldin A (Figure R7, right panel), strongly suggesting that NMI is not secreted by ER-Golgi vesicular secretion pathway.

Figure R7: Left panel, Analysis of endogenous colocalization of NMI in THP1 cells. Nuclei were stained by DAPI. Right panel, NMI in the supernatant(Sup) and cytoplasmic fraction of THP1 cells were analyzed by immunoblotting, pretreated with Brefeldin A for 8 hours and stimulated with *S.typhimurium* for 4 hours.

[Rev. 2.1.3] In fact, the authors state at line 139 of page 6 that "intracellular NMI decreased gradually in both a time- and MOI- dependent manner", but do not show the data. These data are essential.

[Answer to Rev.2.1.3]: We apologize for this unclear description. We have now revised the sentence to "extracellular NMI increased gradually in both a time- and MOI- dependent manner", hopefully to good effect. This result has been incorporated into the revised Figure 2a of the new version of our manuscript.

[Rev. 2.2] 2-important. The authors cannot say (page 5) that "heat pre-treatment... significantly destroyed the activity of NMI protein to about 20%", because destruction is a 100% reduction. In fact, an 80% decrease is not fully convincing, because it suggests that some of the protein refolds after heat treatment, or that the protein preparation contains pyrogens other than LPS (for example cell wall components). I suggest that the authors try digesting the protein preparation with trypsin; if some inflammatory activity remains, they should purify their protein better.

[Answer to Rev. 2.2]: We thank the reviewer for this excellent suggestion. We revised the sentence to "heat pre-treatment... significantly reduced the activity of NMI protein to about 20%". In addition, we performed trypsin treatment and found that the activity of NMI protein was completely destroyed, suggesting that NMI protein but not LPS stimulated TNF α release. This result has been incorporated into the revised Fig. 1d.

[Rev. 2.3] 3-important. The role of caspase 1 is not well investigated, and actually looks minor. Casp1^{-/-} mice still secrete a lot of IFP35 (Fig 5b). This part can actually be omitted completely.

[Answer to Rev. 2.3]: We agree with the reviewer, and have now removed this part from the revised manuscript.

minor:

[Rev. 2.4] - in the introduction, the authors should explain what NMI/IFP35 are and what they do, and what is the phenotype of the Nmi knockout mouse (essential since they use it)

[Answer to Rev. 2.4]: We thank you for the suggestion. We have now revised the introduction in line with the comments. There is no apparent phenotype for NMI knockout mice. To characterize the development phenotype of the innate immune system in the KO mice, we examined the leukocytes in the peripheral blood and the immune populations in spleen and lymph nodes. Our results showed that the types, proportions and numbers of these immune populations were within normal range and comparable with those in WT, suggesting the normal development of the immune system in the KO mice. These new results have now been incorporated into Fig. S4 in the revised manuscript.

[Rev. 2.5]- the initial paper on secreted HMGB1, cited by the authors, used recombinant HMGB1 that was heavily contaminated with LPS. While this does not jeopardize the main conclusion of the paper, the size of the HMGB1 effects were overestimated. In practice, the authors should compare the effect of IFP35/NMI with that of HMGB1 reported in later papers (which is less; the effect of IFP35/NMI actually appears truly large).

[Answer to Rev. 2.5]: We compared the effects of NMI/IFP35 with those of HMGB1, S100A and CIRP reported in recent papers, and found that the proinflammatory activity of NMI was comparable with those DAMPs. The new references have now been added to the revised manuscript.

[Rev. 2.6]- page 4, line 89: NMI is not homologous to IFP35, it may be paralogous (but this has to argued).In practice, it easier to say that NMI and IFP35 share xx% of sequence identity, or belong to the same protein family.

[Answer to Rev. 2.6]: Thank you for this explanation. We have now revised the sentence to “NMI and IFP35 belong to the same protein family”.

[Rev.2.7]- the authors inappropriately use Student's test when they should use one-way ANOVA followed by post-tests between groups. This should not change the overall results, though.

[Answer to Rev. 2.7]: We agree with the reviewer, and apologize for this mistake. We have now repeated our statistical analysis for the revised manuscript.

[Rev. 2.8]- Western blots are very dark and often overexposed to saturation, and there is no mw marker.

[Answer to Rev. 2.8]: We apologize for the inferior quality of our blots. We have now changed all Western Blot figures including Fig. 1e, 2a, 3e, 5a and S6a to improve their quality, and incorporated the new blots into the revised manuscript.

[Rev. 2.9]- IFP35 and NMI can actually form heterodimers. Are the amounts of secreted IFP35 and NMI stoichiometric? Likewise NMI can form hetrodimers with Myc. Is Myc secreted? Either yes or no, this would be interesting to know.

[Answer to Rev. 2.9]: We thank the reviewer for insightful question. Although IFP35 and NMI assemble into high molecular mass complexes in the cytosol, the levels of extracellular IFP35 released by LPS-stimulated RAW264.7 cells were measured as 1-10 ng ml⁻¹, which was approx. 10-fold higher than that of NMI under identical condition (Fig. 2b, 5a). Thus, we would argue that the amount of secreted IFP35 and NMI are not stoichiometric. In addition, stimulated by *S.typhimurium*, Myc protein was not detected in the supernatant (R8), suggesting that Myc was not secreted with NMI.

Figure R8: N-myc in the supernatant (Sup) and cytoplasmic fraction of THP1 cells were analyzed by immunoblotting post *S.typhimurium* stimulation.

Reviewer #3 (Remarks to the Author)

In this submission Xiahou *et al* investigate the possibility that two intracellular interferon stimulated proteins NMI (N-myc interactor, not defined by the authors) and IFP35 can function as DAMPs, stimulating inflammatory responses mediated by Toll-like receptor 4. These are proteins with fairly well defined intracellular functions activating RIGI and forming part of transcription factor complexes. None of this background is discussed. They are not therefore obvious candidates as activators of TLR4, although there is good evidence that other improbable molecules like HMGB1 can act in this way.

Thanks for the reviewer's suggestion. We revised the manuscript and added the intracellular functions of NMI and IFP35 in introduction.

[Rev.3.1] The key issue here is that the NMI and IFP35 proteins have been produced by *E. coli* expression and there is therefore a likelihood of contamination of the preps with LPS, the bona fide TLR4 agonist. They have controlled for this with Limulus assays, treatment with polymyxin and heat treatment of the samples. LPS bound to protein is not detected by Limulus assays or necessarily removed by PM. In my view the authors need to go further to demonstrate the absence of LPS by treating the protein samples with protease and expressing it in an insect or mammalian cell culture system that is free from LPS. As far as heating the sample is concerned this is likely to precipitate the protein along with any bound lipid so is not a good control.

[Answer to Rev.3.1]: Thanks for the constructive suggestion. As shown in Fig. 1d and Fig. S6b, the activities of NMI and IFP35 were abolished after protease treatment. In addition, as shown in Figure S1d and Fig. 5e, the activities of hNMI and hIFP35 were confirmed using purified hNMI and hIFP35 from insect cells.

[Rev.3.2] In Fig 2 they look at release of NMI from cells after treatment with Salmonella, and the hepato-toxin acetaminophen (paracetamol). In all cases these treatments cause cell death so it is not surprising that NMI is released. No other intracellular proteins are assayed (eg LDH) so it is not possible to say that this is a specific release (as has been suggested for HMGB1 1).

[Answer to Rev.3.2]: The intracellular protein LDHB1 (L-lactate dehydrogenase B 1) was used as control in the revised Fig. 2a. The results showed that the release of NMI began at the first 1-3 hours post stimulation, much earlier than the release of HMGB1 and LDHB1 which began at the 5-9 hours. These results suggest that NMI were actively released before it is passively released by dead cells.

[Rev.3.3] For Fig 3 the concerns about LPS apply, further the response to NMI in the HEK293 assay is absolutely tiny. Fig 4 shows that NMI $-/-$ mice are a bit more resistant to LPS induced shock and there is a bit less TNF and IL6 produced. It is important that the genetic backgrounds are the same here, it also unclear to me how this finding relates the proposed activation of TLR4. The final figure recapitulates some of the findings with NMI with IFP35 and uses a truncated protein expressed as GST fusion in *E. coli*, so the same caveats apply about LPS contamination.

In conclusion I think more has to be done to show that these proteins are bona fide activators of TLR4. Also the study does not seem to shed much light on how these proteins are involved in inflammatory pathogenesis.

[Answer to Rev.3.3] The transfected HEK293 cells are not sensitive to stimulation due to low expressions of TLR4/MD2/CD14, the receptors on the cell surface. In previous reports, S100A proteins stimulates a 3-fold increase of NF- κ B activity in a stable transfected luciferase assay in HEK293³. In our paper, the plasmids expressing mTLR4, mMD2, mCD14, and NF- κ B promoter-driven luciferase were transiently transfected into HEK293T. Thus, a 2-fold increase of the NF- κ B activity stimulated by NMI was considered as a significant increase.

In Fig 4, C57BL/6 mice were used both for WT and NMI knockout, as shown in the revised methods. Thus, the genetic backgrounds were same.

We treated NMI and IFP35 using protease as shown in Fig.1d and Fig.S6b in the revised manuscript. In addition, the activities of NMI and IFP35 from insect cell were also tested (Fig. S1d and Fig. 6d). These new results strongly suggest that NMI and IFP35 intrinsically stimulate the release of TNF α , the proinflammatory cytokines. Results in Fig. 3a and Fig. 3c in the revised manuscript showed that TLR4 was necessary for the activity of NMI. In addition, we examined the *in vivo* interactions between TLR4 and NMI (Fig. 3e and Fig. 3d). Thus, we proposed that NMI stimulate inflammation through TLR4 pathway.

Since NMI and IFP35 simulate the release of TNF α , which was both necessary and sufficient to mediate inflammatory pathogenesis, including sepsis. We proposed that NMI and IFP35 serves as proinflammatory DAMPs in the revised manuscript. The functions of NMI and IFP35 in inflammatory pathogenesis still need to be studied in the next step.

Reference

1. Wang H, *et al.* HMG-1 as a late mediator of endotoxin lethality in mice. *Science* **285**, 248-251 (1999).
2. Qiang X, *et al.* Cold-inducible RNA-binding protein (CIRP) triggers inflammatory responses in hemorrhagic shock and sepsis. *Nature medicine* **19**, 1489-1495 (2013).
3. Vogl T, *et al.* Mrp8 and Mrp14 are endogenous activators of Toll-like receptor 4, promoting lethal, endotoxin-induced shock. *Nature medicine* **13**, 1042-1049 (2007).
4. Bertheloot D, Latz E. HMGB1, IL-1alpha, IL-33 and S100 proteins: dual-function alarmins. *Cellular & molecular immunology* **14**, 43-64 (2017).
5. Hara MR, *et al.* A stress response pathway regulates DNA damage through beta2-adrenoreceptors and beta-arrestin-1. *Nature* **477**, 349-353 (2011).
6. Chen J, Shpall RL, Meyerdierks A, Hagemeyer M, Bottger EC, Naumovski L. Interferon-inducible Myc/STAT-interacting protein Nmi associates with IFP 35 into a high molecular mass complex and inhibits proteasome-mediated degradation of IFP 35. *The Journal of biological chemistry* **275**, 36278-36284 (2000).
7. Zhou X, *et al.* Interferon-alpha induces nmi-IFP35 heterodimeric complex formation that is affected by the phosphorylation of IFP35. *The Journal of biological chemistry* **275**, 21364-21371 (2000).

Point-by-point response to the reviewers' comments

We appreciate the reviewers for their critical comments, which greatly helped us to improve the paper. According to these comments, we provided further experimental data and revised the paper including main text and supplementary information.

Reviewers' comments:

Reviewer #1 (Remarks to the Author):

The reviewer notes that the revised version of the manuscript reads significantly better and includes some valuable discussion points as well as improved overall flow. The reviewer requests the following items:

[2-Rev. 1.1] Answer to reviewer 1.2.1 – please include the flow cytometry plots of only dendritic cells recruited to the lymph nodes after LPS treatment.

[Answer to 2-Rev. 1.1] Thanks for the suggestion. The flow cytometry plots of dendritic cells recruited to the lymph nodes after LPS treatment was shown in the following and added in Fig S2.

Figure S2 **The LPS induced septic-shock mice model.** (a) Mobilization of immune populations to lymph nodes after injection of LPS (10mg kg^{-1}) into wild type and knockout mice. Data are presented as the mean \pm s.e.m. of 3 individual mice. (b) Flow cytometric analysis and quantification of dendritic cell (CD11c+CD86+)

populations in lymph nodes.

[2-Rev. 1.2] Answer to reviewer 1.3.1 – please include discussion provided in the rebuttal in the main discussion section of the manuscript. This explanation of extracellular NMI having a distinct function to intracellular NMI is crucial to the mechanism of activation of macrophages by the NMI TLR4 interaction and needs to be clearly stated in the main body of discussion.

The reviewer also suggests that data showing that NMI cannot be taken up into cells after secretion is an important overall finding and should be discussed in more detail in the results as well as discussion section of the manuscript.

[Answer to 2-Rev.1.2] Thank the reviewer for the instructive comment. The distinct functions of extracellular and intracellular NMI have been discussed in the fourth paragraph of the discussion section. The result showing that NMI cannot be taken up into cells after secretion was added in Fig 3a and discussed both in result and discussion sections.

[2-Rev. 1.3] Answer to reviewer 1.6.1 – please include the laminin blot for the cytoplasmic fraction and actin blot for the nuclear fraction to ensure that there was no contamination during the fractionation.

[Answer to 2-Rev. 1.3] Actin is a highly conserved protein that has essential functions in both cytoplasm and nucleus. Nuclear actin is involved in transcription by RNA polymerases, chromatin remodeling, RNA processing, intranuclear transport, nuclear export and in maintenance of the nuclear architecture^{1,2}. Thus, it is not a good control to qualify the fractionation. To evaluate the fractionation, laminB in the cytoplasmic fraction was included in the revised Fig 1e. The result showing that no contamination from nuclear fraction was included in the cytoplasmic fraction.

1 Visa, N. & Percipalle, P. Nuclear functions of actin. *Cold Spring Harbor perspectives in biology* **2**, a000620, doi:10.1101/cshperspect.a000620 (2010).

2 Migocka-Patrzalek, M. *et al.* beta- and gamma-Actins in the nucleus of human melanoma A375 cells. *Histochemistry and cell biology* **144**, 417-428, doi:10.1007/s00418-015-1349-8 (2015).

[2-Rev. 1.4] Answer to 1.8.3. – the reviewer appreciates the authors attempt to show specificity of their antibodies recognizing IFP35 and NMI however, using the knock out cells to validate an antibody when the knock out models themselves are also being validated using the same antibody is circular logic.

Please overexpress the IFP35 and NMI cDNAs and confirm that the antibodies are specific utilizing the appropriate controls.

[Answer to 2-Rev. 1.4] We apologize for the unclear statements of the antibodies used in this paper. The specificity of the antibodies in the ELISA kits were tested and shown in Fig R5. In addition to the ELISA kits, we used three different antibodies including Anti-hNMI (ab183724), Anti-mNMI and Anti-hIFP35 (H00003430-D01P). The Anti-hNMI (ab183724) used in Fig 2a and Fig 3f was purchased from Abcam.

The Anti-mNMI used in Fig S3 was produced using recombinant mNMI according to previous report³. The Anti-hIFP35 (H00003430-D01P) used in Fig 5a was purchased from Abnova. Since hIFP35 and mIFP35 are highly conserved, Anti-hIFP35 recognizes both mIFP35 and hIFP35. Thus, Anti-hIFP35 was also used in Fig S3 to evaluate the knockout mice model of IFP35.

According to the reviewer's suggestion, we expressed mIFP35, hNMI and mNMI in HEK293 cells. The specificity of the antibodies was tested using non-transfected cells and recombinant protein as negative and positive control, respectively. The result is shown in Fig S6a, which is combined with previous R6 in the revised Fig S6.

Figure S6: The specificity of the antibodies and ELISA kits used in the paper. (a) HEK293T cells were transfected with hNMI, mIFP35 or mNMI and the specificity of Anti-hNMI (ab183724), Anti-hIFP35 (H00003430-D01P, also used for mIFP35) and Anti-mNMI were assessed by immunoblotting. (b-c) The specificity of the ELISA kits. Panel b shows the ELISA analysis of IFP35 and NMI in the serum of WT, *mNmi*^{-/-} or *mIfp35*^{-/-} BMDM mice, treated with LPS (10 mg kg⁻¹). Panel c shows the proteins bound to antibody in the ELISA kits, assessed by immunoblotting.

- Leenaars M, Hendriksen CF. Critical steps in the production of polyclonal and monoclonal antibodies: evaluation and recommendations. *ILAR journal* **46**, 269-279 (2005).

Reviewer #2 (Remarks to the Author):

The authors have replied to my queries and the other reviewers' in a complete and satisfactory way. Specifically, the absence of response to NMI and IFP35 proteins digested with trypsin indicates that there is little or no contamination from LPS and other PAMPs.

I still have some technical issues though.

[2-Rev. 2.1] Some Western blots are still overexposed to the point that they cannot be properly evaluated. The offending ones are in Fig 2a (the lysate part) and Fig 3e.

[Answer to 2-Rev. 2.1] Fig 2a and Fig 3e have been further revised in the newly revised paper.

[2-Rev. 2.2] The statistical test used in Figure 2e should be indicated. Obviously, it must be a Mann-Whitney test. The correlation between patient outcome (dead/alive) and serum NMI appears stronger than the authors actually hint: the authors could apply a simple chi-square test (which turns out highly significant) or a more sophisticated ROC model.

[Answer to 2-Rev. 2.2] We appreciate the reviewer for his/her suggestion. A Mann-Whitney test was used in Figure 2e and indicated in the figure legend.

[2-Rev 2.2] Figure R5 should be added to the paper as a supplementary figure.

[Answer to 2-Rev 2.2] Figure R5 was added to the paper as Fig S6b. In addition, we tested the specificity of the antibodies used in the paper, as shown in Fig S6a.

[2-Rev 2.3] Figure R7 should be added as a new panel of Fig. 2 and appropriately described in the main text

[Answer to 2-Rev 2.3] Since the locations of intracellular NMI have been reported. We only added the right panel of Figure R7 as Fig. 2b in the main text.

[2-Rev 2.4] Figure R8 should be added as a new panel to Fig 2 as a very convincing negative control.

[Answer to 2-Rev 2.4] Figure R8 was involved in Fig 2a in the revised paper.

[2-Rev 2.5] Conversely, some supplementary figures should be shown in the main text or dispensed with

Fig S2b should be added as a new panel to Fig 2, Fig 2a can be omitted altogether.

Fig S3a,b can be omitted altogether. The experiment in Fig S3c is not convincing, and should be omitted altogether. Fig S3d should be added as new panel to Fig 3, documenting physical interaction.

[Answer to 2-Rev 2.5] In the revised paper, Fig S2b and S3d are shown as Fig. 2c and Fig. 3f, respectively. Fig S2a, S3a, S3b and S3c are omitted.

[2-Rev 2.6] Minor: gene names for mice are indicated with the first letter capitalized and the subsequent ones in NOT capital

[Answer to 2-Rev 2.6] Thanks for the comments. We apologize for the mistake in the original manuscript. The gene names have been revised.

Reviewer #3 (Remarks to the Author):

[2-Rev 3.1] They appear to have addressed most of the points that I made with new experimental data that on the face of it looks to be of good quality. The only substantive remark I have is that they should have had a control where the LPS was treated with protease (Fig. 1D, S6).

[Answer to 2-Rev 3.1] We thank the reviewer for the critical comment. The LPS controls treated with protease have been added into the revised Fig. 1d and S5b, the previous S6b.

Point-by-point response to the reviewers' comments

We appreciate the reviewer for their critical comments. According to these comments, we further revised the paper in the discussion section.

Reviewer #1 (Remarks to the Author):

This version of the manuscript is much superior to all the previous versions. Certainly, most issues that were raised have been addressed. However, the manuscript so far fails to clarify the following 3 things which this reviewer thinks are fundamentally germane to the premise and addition of these conceptual clarification will complete the manuscript in all aspects.

[3-Rev. 1.1] The findings described state that secretion mechanism of NMI is ER-golgi independent (Figure 2b), it fails to elucidate which is the actual mechanism of secretion.

[Answer to 3-Rev. 1.1] We thank the reviewer for the suggestion. The secretion mechanisms of many DAMPs are elusive and not clear till now. Since the release of NMI is independent the two classic pathways, ER-Golgi and proptosis, it is hard to clarify its secretion mechanism at this stage because it may depend on some modifications or unknown release pathways. This question is discussed in the 5th paragraph of discussion section.

[3-Rev. 1.2] Figure 2 states that NMI is released by activated macrophages and Figure 3 states that NMI stimulates macrophages. Thus, there appears to be a circular nature of signaling. Does this mean that the 1st secretion of these DAMPs occur through activated macrophages? If so, a diagrammatic representation of the proposed model of the action of these DAMPs will be highly desirable as a part of the published manuscript.

[Answer to 3-Rev. 1.2] Thanks for the suggestion. In this paper, we didn't declare that NMI is released by NMI-stimulated macrophage, although we show that it is released by LPS-stimulated macrophage. Thus, the possible circular nature of signaling need to be further studied. In addition, we are not sure if NMI is the first DAMP released by LPS-stimulated macrophage although it is released much earlier than HMGB1. Based the known knowledge, the 1st released DAMP should dependent on the infection or injury states of the tissue. Thus, we didn't propose a circular model for NMI and IFP35.

[3-Rev. 1.3] Overall the manuscript has more details about NMI compared to IFP35. It is highly recommended to present additional supporting data is presented to make a more compelling case for IFP35 having the identical behavior. For example, it remains unclear if the secretion mechanism (or lack of it) discussed in Figure 2b is true about IFP35. It is also unclear if Figure 2 and Figure 3 are applicable to (extrapolated) statements made for IFP35. Additional data to support the claims of the manuscript that equate NMI and IFP 35 will fulfill this lacuna.

[Answer to 3-Rev. 1.3] In this paper, we propose that NMI and IFP35 serve as DAMPs. Since the secretion pathway is not clear, we didn't claim that NMI and IFP35 are equated. Thus, we discussed that "although both trigger the activation of macrophage and exacerbate inflammation, the mechanisms of secretion and immune regulation of these two proteins may be different. The differences and possible crosstalk between IFP35 and NMI during inflammation need to be studied in the future."

Reviewer #2 (Remarks to the Author):

The authors have satisfied all my queries, and I have no further comment.
Congratulations for a well-done job.

Reviewer #3 (Remarks to the Author):

In this second revision the outstanding point I made has been addressed.